# Technical note: Determining Arctic Ocean halocline and cold halostad depths based on vertical stability

Enrico P. Metzner[1] and Marc Salzmann[1]

[1]Institute for Meteorology, Universität Leipzig, Leipzig, Germany,

**Correspondence:** Enrico P. Metzner (enrico.metzner@uni-leipzig.de)

**Abstract.** The Arctic Ocean halocline separates the cold surface mixed layer from the underlying warm Atlantic Water (AW), and thus provides a precondition for sea ice formation. Here, we introduce a new method in which the halocline base depth is diagnosed from vertical stability and compare it to two existing methods. We also propose a novel method for detecting the cold halostad, a layer characterized by a small vertical salinity gradient, which is formed by Pacific Winter Water in the Canada Basin or by melt water off the eastern coast of Greenland and also Svalbard. Our main motivation for diagnosing the halocline base depth depending on vertical stability was that vertical stability is closely related to vertical mixing and heat exchange. Vertical stability is a crucial parameter for determining whether the halocline can prevent vertical heat exchange and protect sea ice from warm AW. When applied to measurements from ice-tethered profilers, ships, and moorings, the new method for estimating the halocline base depth provides robust results with few artifacts. Analyzing a case in which water previously homogenized by winter convection was capped by fresh water at the surface suggests that the new method captured the beginning of new halocline formation in the Eurasian Basin. Comparatively large differences between the methods for detecting the halocline base depth were found in warm AW inflow regions for which climate models predict halocline thinning and increased net surface energy fluxes from the ocean to the atmosphere.

## 1 Introduction

The Arctic Ocean outside the main Atlantic warm water inflow regions and the shallow marginal shelf seas is usually stratified into a cold and fresh surface mixed layer (SML), which is from $\sim$5 to >100 m thick, depending on region and season (Peralta-Ferriz and Woodgate, 2015), a halocline below the SML with a base depth $\sim$40 to >200 m (Fig. 4 of Polyakov et al., 2018), a layer of warm and saline Atlantic Water (AW) below the halocline centered near 300 to 500 m in the Eurasian Basin and somewhat deeper in the Canada Basin (Aagaard et al., 1981; Macdonald et al., 2015), and deep water below. Convection in the SML is driven by surface cooling and brine release during sea ice formation, with maximum SML depth in winter. River inflow and precipitation act as sources of fresh water. Below the SML, salinity increases in the halocline. Within the halocline, one can distinguish between the cold halocline layer (CHL) in the Eurasian Basin, the Pacific Halocline Waters (PHW, modified Pacific Water which originally entered the western Arctic via the Bering Strait) in the Amerasian Basin, and the lower halocline waters (LHW, water of Atlantic origin which is less modified compared to CHL water) (e.g. Alkire et al., 2017; Polyakov et al., 2018; Anderson et al., 2013). In the CHL, the temperature remains close to the freezing point. Several processes have been

suggested as contributors to LHW and CHL formation. Based on data from the *Oden* 1991 cruise, Rudels et al. (1996) found that new halocline formation was initiated by the advection of relatively fresh shelf waters near the surface above denser and more saline water below, when the advection of the fresh water limited winter convection. Alkire et al. (2017) and Rudels et al. (2004) argued that convective homogenization and capping by fresh water due to sea ice melting in the inflow from the Fram Strait and the Barents Sea can transform AW into halocline water. Rudels et al. (2004) stressed that melting provides a precondition for sea ice formation and convective homogenization during the following winter. Another process which has been widely discussed, and which is thought to be especially important for the PHW is the advection of dense and saline shelf waters (where salinity increases due to brine release during sea ice formation especially in winter) below the SML (Aagaard et al., 1981; Jones and Anderson, 1986; Rudels et al., 2004). While halocline formation via convective homogenization and capping does not require dense shelf waters, convective homogenization and capping can also occur after (i.e. in addition to) the advection of dense shelf water (Steele and Boyd, 1998; Rudels et al., 2004). The PHW in the Canada Basin originates from Pacific Water inflow, which is modified on the Chukchi Sea Shelf, while the LHW is of Atlantic origin not only in the Eurasian Basin but also in the Canada Basin (e.g. Anderson et al., 2013). Because of seasonal modifications on the Chukchi Sea Shelf, the PHW in the Canada Basin can be further subdivided into Pacific Winter Water (PWW) and less saline and warmer Pacific Summer Water (PSW) (e.g. Timmermans et al., 2014). Compared to the CHL in the Eurasian Basin, in the PWW, the salinity is lower and the salinity gradient is smaller. This is why Shimada et al. (2005) called the layer which is formed by PWW a cold halostad (CHS). Below, we argue that an increase in salinity associated with the PHW and another increase associated with the LHW results in two distinct local stability maxima between the surface and the LHW (compare also schematic in Supplement 1 and profiles in Supplement 2). The lower one of these two stability maxima is absent in the presence of a CHL in the Eurasian Basin (except in regions off the eastern coast of Greenland and also Svalbard). This allows us to identify the CHS. North east of Greenland glacial melt water forms an intermediate low salinity layer with small salinity gradient which is also called a cold halostad (Dmitrenko et al., 2017).

Because density is more influenced by salinity than temperature if the temperature is low (Aagaard et al., 1981; Roquet et al., 2022) a configuration with warm AW underlying colder halocline water is stable. The presence of a (cold) halocline thus insulates the SML from direct contact with the warm AW and protects sea ice from the warm AW (Aagaard et al., 1981; Lind et al., 2016; Polyakov et al., 2017, 2020). Conversely, a retreat of the CHL in the Eurasian Basin leads to increased vertical mixing as observed and described by Steele and Boyd (1998); Björk et al. (2002); Polyakov et al. (2017). Retreating sea ice, increased surface heat flux and the retreat of the halocline have been called atlantification of the Eurasian Basin (Polyakov et al., 2017). Future climate model projections for a high emission scenario also showed very large temperature gradients directly below the surface mixed layer more frequently, especially during the cold season. The associated heating of the SML in combination with sea ice loss resulted in further increased annual mean upward net surface energy fluxes outside the Central Arctic along the main warm water inflow pathways (Metzner et al., 2020). Therefore, consistent and robust descriptions of the halocline and cold halostad layer boundaries are important to understand the evolution of the structure of the upper Arctic Ocean in the past and the future. While the halocline generally protects sea ice, PSW can be warm enough to participate in sea ice melting (e.g. Shimada et al., 2006; Timmermans et al., 2014).

Several methods have been proposed for identifying the halocline based on observations. Steele et al. (1995) identified cold halocline water based on conditions for salinity ($34 < S < 34.5$ in the practical salinity scale) and temperature (T<-0.5°C). Rudels et al. (1996) defined the boundaries of the CHL by using the 34.3 isohaline. Bourgain and Gascard (2011) used a density ratio threshold to define the base of the halocline. The density ratio is the ratio of temperature and salinity contributions to the vertical stability. A large density ratio implies that the vertical stratification is dominated by temperature and a small density ratio implies that stratification is dominated by salinity. The density ratio threshold suggested by Bourgain and Gascard (2011) assumes that oceanic layers above the halocline base are almost entirely salt-stratified with temperature contributing less than 5% to the total stratification (Polyakov et al., 2018). This density ratio method was adopted among others by Polyakov et al. (2017, 2018) and Metzner et al. (2020). Using tracer observations in the western Eurasian Basin, Bertosio et al. (2020) found the base of the LHW to be located at a density of $1027.85\,\mathrm{kg\,m^{-3}}$. Analyzing salinity and temperature observations from the Makarov Basin and along the East Siberian continental slope, Bertosio et al. (2022) again defined the base of the halocline using a density threshold and compared the results obtained with this definition to those obtained with other definitions from the literature. A fairly simple and robust method for computing the CHL base depth was proposed by Metzner et al. (2020). In this method, the base of the CHL is determined by a temperature difference of $1\,°\mathrm{C}$ between water temperature and its freezing temperature. This temperature difference method is very sensitive to warming from below, while the density-ratio method of Bourgain and Gascard (2011) is very sensitive to the salinity profile. One drawback of the temperature difference method is a potential dependence of the optimal threshold value on region (Metzner et al., 2020). Polyakov et al. (2018) proposed an indicator of the potential of the Arctic halocline to prevent vertical mixing based on available potential energy, adapting the density ratio threshold of Bourgain and Gascard (2011) to identify the halocline base.

Here, we propose a new method to identify the halocline base using a vertical stability threshold and compare it to two existing methods using measurements from ice-tethered profilers, ships, and moorings. Our main objective was to devise a method that uses a threshold value of a variable which is closely related to the role of the halocline in insulating the SML from the warm AW. The choice of a vertical stability threshold was motivated by the argument that vertical stability is more directly related to vertical mixing than either density, temperature, or the density ratio. Our second goal was to devise a particularly robust method to detect the halocline base. Based on the argument that the presence of PWW forming a CHS on top of LHW creates a stability profile with two distinct local stability maxima, we also propose a method for estimating the boundaries and the center of the CHS.

In the next section, we describe methods to determine the halocline base depth, a method for estimating the CHS upper and lower boundaries and the CHS center, and introduce observational datasets used for comparison and testing. In Sect. 3, we compare the new stability method for determining the halocline base depth to two existing methods and test the new method for determining the CHS depth and extent. The results are summarized and discussed in Sect. 4.

## 2 Methods and Data

### 2.1 Methods for estimating the halocline base depth

#### 2.1.1 Density ratio (DR) method

In the Arctic halocline, the density gradient due to temperature is small compared to the density gradient due to salinity by definition (Bourgain and Gascard, 2011). The density ratio (DR) method by Bourgain and Gascard (2011) therefore identifies the halocline base by the requirement that the ratio $R_\rho$ between the density gradient due to temperature and the density gradient due to salinity must remain below a certain threshold. The density ratio is defined as $R_\rho = (\alpha \nabla_z \theta) / (\beta \nabla_z S)$ with potential temperature $\theta$ in °C, salinity $S$ in the practical salinity scale and depth $z$ in m. $\alpha = -\rho^{-1}(\partial\rho/\partial\theta)$ and $\beta = \rho^{-1}(\partial\rho/\partial S)$ are the thermal expansion coefficient and the haline contraction coefficient, respectively. Bourgain and Gascard (2011) empirically estimated that searching downward for the depth, in which $R_\rho$ exceeds $0.05$, provides a reasonable estimate for the base of the halocline. The search starts at the base of the SML (here determined as described in Sect. 2.3 below), which is defined to be the top of the halocline layer. If the density ratio threshold is exceeded already directly at the base of the SML, then no halocline was detected for the corresponding profile. Such profiles are excluded when computing statistics of halocline base depths. Similarly to Bourgain and Gascard (2011), we smoothed the $S$ and $\theta$ prior to computing the density ratio as explained in Sect. 2.4.

#### 2.1.2 Temperature difference (TD) method

The temperature difference (TD) method (Metzner et al., 2020) uses the difference $\Delta T$ between the ocean temperature $T$ and the sea water freezing temperature $T_\mathrm{f}$ to estimate the cold halocline base depth. The freezing temperature was calculated from Gill (1982). Searching downward, starting at the SML base, the base of the halocline was calculated as the depth, in which $\Delta T$ first exceeds $1\,\mathrm{K}$. This threshold was estimated to be high enough, that the "cold" core of the cold halocline layer is detectable, and low enough to separate the CHL from the AW with core temperature approximately $1.5\,°\mathrm{C}$ to $3\,°\mathrm{C}$ ($T_\mathrm{f} \approx -2\,°\mathrm{C}$ leads to $\Delta T \approx 3.5\,°\mathrm{C}$ to $5\,°\mathrm{C}$ at the core). In cases in which the temperature threshold was first exceeded in a depth shallower than $80\,\mathrm{m}$, the search was continued below this depth. If the temperature threshold was exceeded already at the SML base, no halocline was detected. The algorithm was applied to smoothed temperature data (see Sect. 2.4).

#### 2.1.3 Stability (ST) method

The new stability (ST) method prescribes a threshold for the local vertical stability in order to estimate the halocline base depth. Vertical stability is more closely related to vertical mixing than either the density ratio or the temperature difference. However, while vertical stability is closely related to vertical heat exchange and to the role of the halocline for protecting sea ice, the density ratio is more directly related to the original definition of a halocline.

Because the search direction was found to affect the robustness of the method, and because stability is decreasing with depth between the AW and the core of the halocline, we search upward instead of downward for the stability threshold. The stability

was computed from $L = \log_{10}(N^2)$, where $N = \sqrt{-(g/\rho)(\partial \rho/\partial z)}$ is the Brunt-Väisälä-frequency (with density computed from pressure, smoothed $S$, and $\theta$ as described below). The stability threshold was approximated based on the density ratio threshold $R_\rho = 0.05$ assuming an approximately constant salinity gradient near the halocline base in the LHW. It was derived starting from the following relationship:

$$\rho^{-1}\nabla_z\rho = \beta\nabla_z S - \alpha\nabla_z\theta = \beta\nabla_z S(1 - R_\rho) \tag{1}$$

With stable $\beta = (7.82 \pm 0.03) \cdot 10^{-4}$ over a wide range of temperature, salinity and pressure values ($-1.2\ldots2.0\,°\mathrm{C}$, $32\ldots37$, $50\ldots350\,\mathrm{dbar}$) and the salinity gradient in $\mathrm{m}^{-1}$:

$$L = \log_{10}(-\nabla_z S) - 2.137 \pm 0.002 \tag{2}$$

Expecting the salinity gradient to be around $0.01\,\mathrm{m}^{-1}$ near the base of the halocline, the resulting stability threshold should be $L \approx -4.14$.

This threshold is searched from $600\,\mathrm{m}$ or at the lowest point (at least $500\,\mathrm{m}$ deep outside the shallower regions according to the conditions for including profiles in the analyses described below in Sect. 2.4) to the surface, as no CHL base was observed deeper than that. Seldom, the first estimate is in warm AW at $T > 0\,°\mathrm{C}$. In such cases, a second search for the stability threshold is started slightly above. If the stability threshold is never exceeded or only exceeded where $T > 0\,°\mathrm{C}$ in a given profile, then no halocline base was detected for this profile.

## 2.2 Cold halostad (CHS) boundary and center estimates

A CHS is formed by PWW in the Canada Basin and also by melt water off the eastern coast of Greenland and Svalbard. Compared to the CHL in the Eurasian Basin, a CHS is characterized by a smaller salinity gradient because of the different water origins. As demonstrated below in Sect. 3.3, this leads to one local stability maximum above the the CHS (at the transition between SML and PHW) and a second stability maximum associated with the transition between PHW and LHW (compare also Supplement 1 and 2). The stability minimum between these two local stability maxima is associated to the CHS. Therefore, as a first condition for identifying a CHS, we require that more than one local stability maximum must be present between the base of the SML and the base of the halocline as identified by the ST algorithm described above. Because the stability profiles computed from temperature and salinity observations contain small scale fluctuations even after smoothing the $S$ and $\theta$ data that is used for computing density as described below, we identify local maxima by first computing a "moving" stability maximum for a $50\,\mathrm{m}$ vertical box surrounding each observation. This moving stability maximum is computed from $L_m(z) = \max(L(z')$ for $|z' - z| < 25\,\mathrm{m})$, where $z$ is depth. Please refer to Supplement 3 for an example illustrating the entire procedure for estimating the CHS boundaries and center depth. The moving maximum operation is illustrated by an example in Fig S3b in Supplement 3. This moving maximum operation was defined in analogy to a moving average. The result is a profile of stability maxima $L_m(z)$ with few local maxima. We then compute the mean of the deeper stability maximum, which is associated with the transition between PHW and LHW, and the stability minimum between the upper and the lower stability maximum, which is associated with the CHS based on the original smoothed stability profile (Supplement 3). This value is

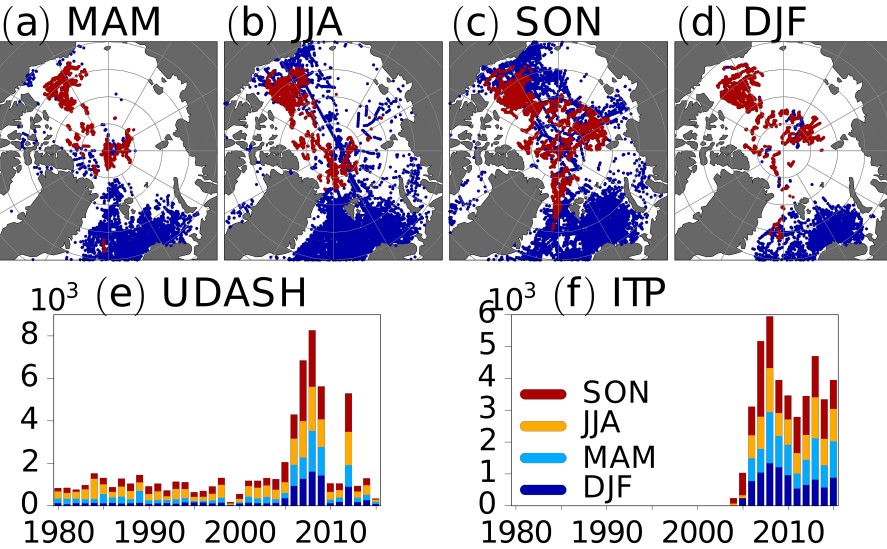

**Figure 1.** Locations of observations for each season (starting with MAM for March, April, and May) with blue dots for UDASH profiles and red dots for ITP profiles (**a–d**). Temporal coverage for UDASH (**e**) and ITP (**f**) observations.

used as a threshold to define the upper and the lower boundary of the CHS. The depth of the center of the CHS is defined as the mean of the upper and lower boundary of the CHS. A CHS will only be recognized by the algorithm if the vertical distance between the deeper stability maximum and the first upper occurrence of that same stability value is at least 50 m, and the difference of $L$ between the lower stability maximum and the local minimum in the CHS is at least 0.2. With this definition, we never identified more than a single CHS per profile.

## 2.3 SML depth estimate

The SML depth was estimated by a change in potential density of $0.125\,\mathrm{kg\,m^{-3}}$ at the surface as in Polyakov et al. (2017). In cases, in which a CHL is detected, the depth of the SML corresponds to the top of the CHL. Potential density was computed from smoothed $S$ and $\theta$. The smoothing was performed using a Gaussian filter as described in the next section.

## 2.4 Data and preprocessing

Temperature and salinity observations were taken from the ice tethered profiler (ITP) project (Krishfield et al., 2008; Toole et al., 2011) and the Unified Database for Arctic and Subarctic Hydrography (UDASH, Behrendt et al., 2018). The ITPs measured temperature, salinity, and pressure twice a day while drifting with the ice floe they were tethered to. Data processing for the ITP data is described by Krishfield et al. (http://www.whoi.edu/fileserver.do?id=35803&pt=2&p=41486). Here, we used processed ITP Level III data. Producing Level III data included removal of corrupted data, corrections for the sensor response behavior, calibrations, and final screening of spurious outliers. ITPs deployed in the Arctic Ocean before 2018 were included here. The

vertical resolution for ITP level III data is 1±0.1 dbar. The accuracy of the sensors used for the ITP observations is 0.002 °C for temperature and 0.002 for salinity according to the manufacturer (Janzen et al., 2016). For temperature, this accuracy range is supported by Wong et al. (2023a). For salinity, larger biases can arise due to sensor shift on longer time scales, depending on the manufacturing date of the sensor (Wong et al., 2023b). The UDASH data set contains data from ships, ice-tethered profilers, profiling floats and other platforms (Behrendt et al., 2018). Only profiles, for which both temperature and salinity were available, were analyzed here. Furthermore, only profiles with a vertical resolution finer than 2.5 dbar in the upper 300 m and a vertical resolution finer than 5 dbar elsewhere were used. We also required that the deepest point in a profile must reach at least 500 m or 90% of the basin depth. This choice addresses the issue of potential sampling biases due to limited vertical extent of the observed profiles. Regions shallower than 100 m were always excluded from the analysis. Bathymetry data was taken from the General Bathymetric Chart of the Oceans (GEBCO) dataset (GEBCO Bathymetric Compilation Group 2021, 2021). This filtering left a total of 43715 ITP and 62012 UDASH profiles. Figure 1 provides an overview of the spatio-temporal coverage of the data. Most measurements are concentrated in the Barents Sea and only few were taken in the Central Arctic during winter. For the East Siberian Sea and the interior of the Laptev Sea, no data was available for winter and spring. Salinity was given in the practical salinity scale.

Depth was computed from pressure using the hydrostatic equation. Density was computed based on salinity, temperature, and pressure. In order to reduce noise, $S$, $T$, and/or $\theta$ were smoothed using a standard one-dimensional Gaussian filter (convolution with a Gaussian function, e.g. Deng and Cahill, 1993) with a standard deviation of 2 dbar and a truncation at ±10 dbar. When using thresholds to estimate the SML or CHL base depth, variables were linearly interpolated between two adjacent depths. Consequently, the SML or CHL base can be located between two vertical observation points and the SML and CHL base depths do not necessarily have to coincide with the depths of the observations.

## 3 Results

### 3.1 Comparison of methods for deriving halocline base depth using case studies

Figure 2 compares three different methods for determining the halocline base depth for ITP-74. Starting from the Laptev Sea in September 2013, ITP-74 drifted across the Central Arctic, almost reaching the East Greenland Sea (Fig. 2a). Until May 2014, Fig. 2b–d shows evidence of a well-defined and stably stratified CHL below the SML. The vertical stratification observed by ITP-74 prior to May 2014 and the performance of the three methods for determining the halocline base depth are further analyzed in an individual profile from this period in Fig. 3. Supplement 4 shows a corresponding figure based on Level I data instead of the more processed Level III data and equations from McDougall et al. (2010) instead of Gill (1982). Supplement 5 shows the locations of the profiles analyzed in Fig. 3 together with maps of sea ice concentration based on Spreen et al. (2008).

Individual profiles of salinity and temperature for 13 January 2014 in Fig. 3a show the base of the SML at about ∼30 m. Between the SML base and ∼80 m a strong salinity gradient and temperatures close to the freezing point indicate a well defined CHL, which is ∼60 m thick. Between the CHL and the AW, temperature and salinity increase in the LHW. Below ∼170 m warm and saline AW is found (please note the kink in the temperature and salinity profiles at ∼170 m in Fig. 3a). The

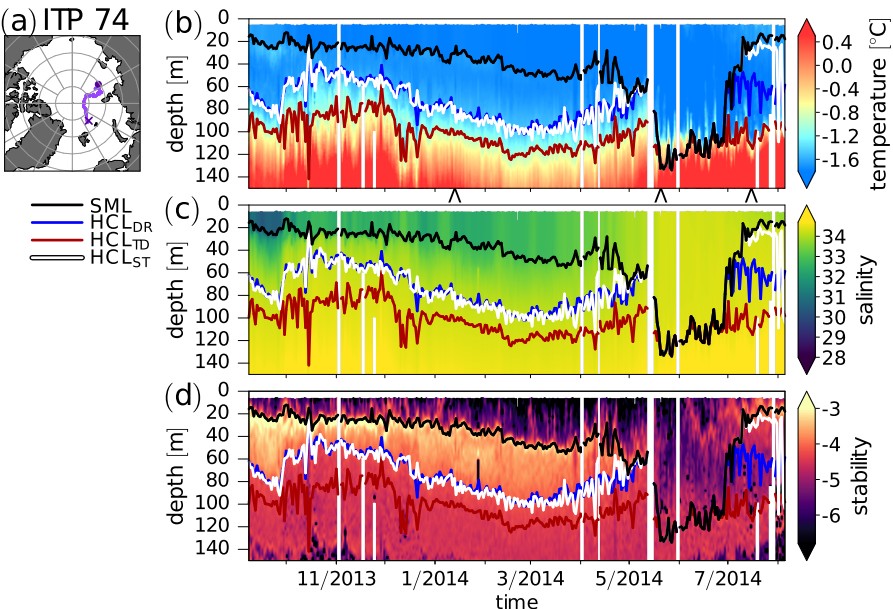

**Figure 2.** ITP-74 location of measurements **(a)** and time series of temperature **(b)**, salinity (in the practical salinity scale) **(c)**, and vertical stability ($L$, unitless, see Sect. 2.1.3) **(d)**. The circle and the cross in (a) mark the beginning and the end of the ITP-74 track, respectively. The colored lines in (b–d) are the base of the SML (black) and the halocline (HCL) base depths derived by the DR method (blue), the TD method (red) and the ST method (white). Individual profiles at the location of the wedge symbols (∧) below the x-axis in (b) are shown in Fig. 3. Profiles that started below 15 m were excluded (only) in this figure (but nowhere else), because this increased readability by reducing the effect of noise in determining the SML base without affecting the overall result.

potential density anomaly used to compute the SML base is shown in Fig.3b. Figs 3c-e show the density ratio, the temperature
difference, and the stability for the observations on 13 January 2014. Threshold values used to identify the halocline base with the DR, the TD, and the ST method are also shown. For the profile observed on 13 January 2014, the DR method (Fig. 3c) and the ST method (Fig. 3e) identify the CHL base, while the TD method places the halocline base in the LHW, somewhere between CHL and AW (Fig. 3c). The stability profile in Fig. 3d yields distinctly different stabilities for the SML, the CHL, the LHW, and the AW. A caveat is that observations are known to show spurious salinity extremes in high-gradient segments
because of different sensitivity and time constants of temperature and conductivity sensors (e.g. Johnson et al., 2007). This can have consequences in the application of the methods. For example, the DR method may select density ratio values affected by the spurious salinity extremes. Although here Level 3 ITP data was used, which is processed data applying sensor corrections, it is possible that salinity profiles exhibit false extremes in areas with significant temperature and salinity gradients.

     In May 2014, the SML deepens and the CHL disappears (Figure 2), as previously noted by Polyakov et al. (2017). During
this convection event, neither of the three methods identified a halocline. Figs 3h–j show profiles for 20 May 2014 after the onset of convection. On this date, the threshold for identifying the halocline base was already exceeded at the SML base for the DR and the TD method (Fig. 3h and i), while the threshold was not reached for the ST method (Fig. 3j). In July, the

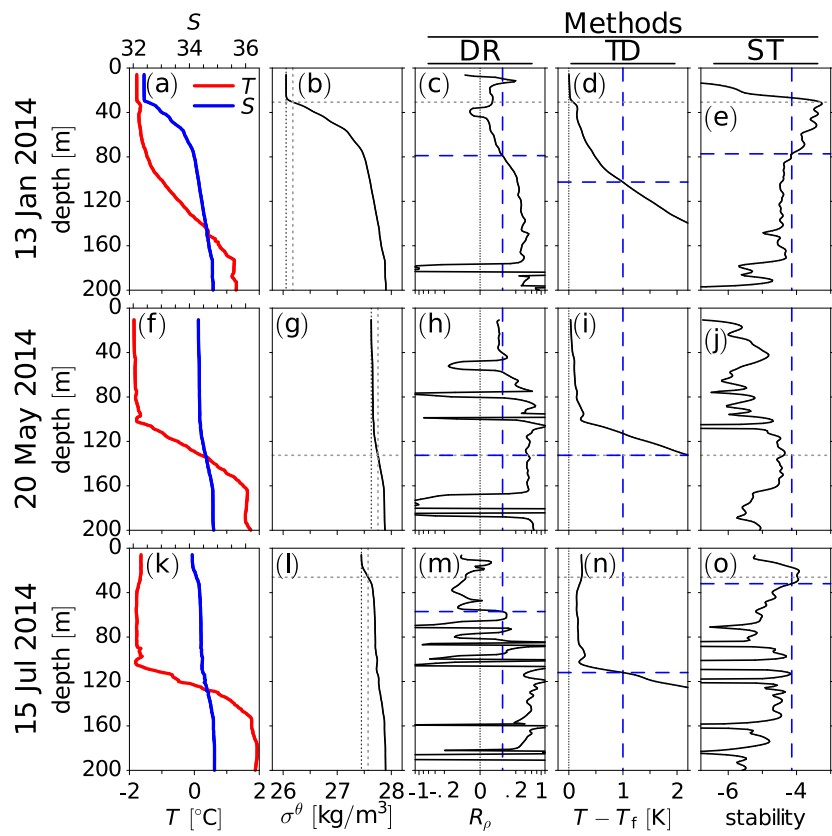

**Figure 3.** Temperature $T$ and salinity $S$ **(a, f, k)**, potential density anomaly $\sigma^\theta = \rho^\theta - 1000\,\mathrm{kg\,m^{-3}}$, where $\rho^\theta$ is potential density **(b, g, l)**, density ratio $R_\rho$ (please note the cubic scaling of the x-axis) **(c,h,m)**, temperature difference $T - T_\mathrm{f}$, where $T_\mathrm{f}$ is the freezing temperature **(d, i, n)**, and stability $L$ **(e, j, o)** from ITP-74 before winter convection on 13 January 2014 (a–e), during winter convection on 20 May 2014 (f–j) and after winter convection on 15 July 2014 (k–o). Vertical dotted lines in b, g, l indicate the surface density anomaly and the threshold for determining the SML base (i.e. surface density plus $0.125\,\mathrm{kg\,m^{-3}}$). Horizontal dotted lines indicate the SML base. Vertical dashed lines indicate threshold values for determining the halocline base. Horizontal dashed lines indicate the halocline base determined by the three different methods. Dashed lines overlying dotted lines in h and i indicate that a threshold for identifying the halocline base was exceeded at the SML base.

situation becomes particularly interesting. The stability at about 80 m depth remains low, pointing to the residual of a mixed layer well below the diagnosed SML base (Figure 2). Fig. 3k for 15 July 2014 (after convection) also shows freshening and warming near the surface. This indicates that relatively fresh melt or/and shelf water may have been advected above a colder and saline layer, which had previously been homogenized by winter convection. The freshening near the surface leads to a salinity gradient below, and also a stability maximum, which is captured by the ST method (Fig. 3o). This appears to be consistent with the convective homogenization and capping mechanism for halocline formation described by Rudels et al. (1996). Figure 2 suggests that in this particular case, the convection may have affected halocline water because prior to the

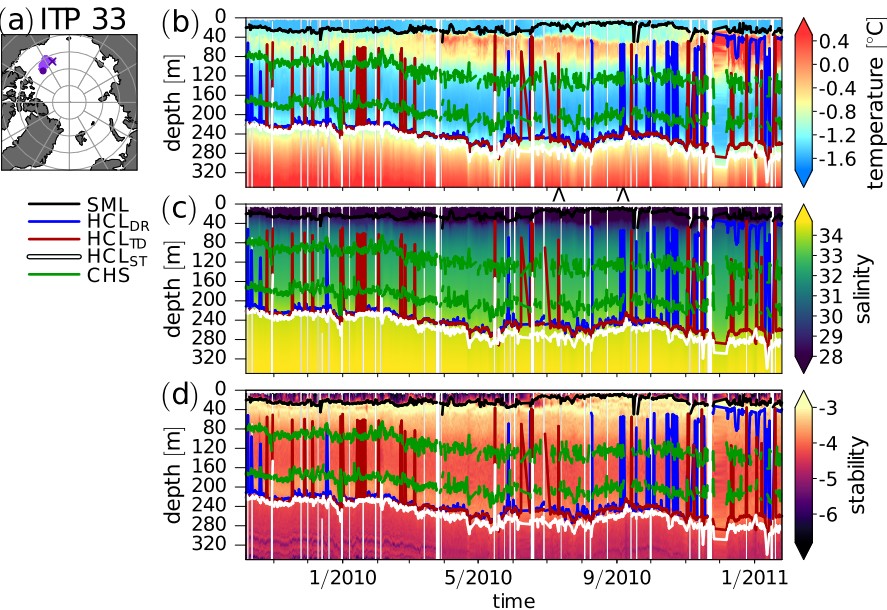

**Figure 4.** As Fig. 2 but for ITP-33. Additionally, the results of the cold halostad bound estimation are shown in dark green.

onset of convection, Fig. 2 shows a well-defined halocline. This is also consistent with a study by Steele and Boyd (1998), who suggested that the winter convection and capping mechanism can act in addition to the advection of dense and saline shelf waters. In the Steele and Boyd (1998) mechanism, which combines findings by Rudels et al. (1996) with earlier findings (e.g. Aagaard et al., 1981), high salinity in the capped water derives from advected cold and dense shelf water, which may previously have been affected by brine release in shelf seas, and not directly from AW. Here, the origin of the halocline water is unclear.

However, Fig. 3o for 15 July 2014 (after winter convection) identifies a stability maximum associated with fresh water near the surface. This suggests that the ST method might indeed be useful for identifying the beginning of new halocline formation via the Rudels et al. (1996, 2004) convective homogenization and capping mechanism or the Steele and Boyd (1998) mechanism which essentially assumes that the Rudels et al. (1996) convective homogenization and capping mechanism acts in addition to the advection of dense shelf water (e.g. Aagaard et al., 1981).

Figure 4 again compares the three different methods for determining the halocline base depth, but this time for ITP-33, which drifted in the Canada Basin between October 2009 and January 2011, where it encountered PSW on top of PWW. In addition to the halocline base depth, Fig. 4b–d shows the CHS boundaries which have been estimated based on stability as described above. The performance of the new algorithm for identifying the CHS will be discussed further below. For now, the main focus will be on isolated spurious minima of the halocline base depth. As evidenced by the spikes in Fig. 4b–d, such

isolated spurious base depth minima occur for the DR and the TD algorithm, but not the ST algorithm. Figure 5a–e shows a case from ITP-33 on 11 July 2010 in which the TD method produced an isolated halocline base depth minimum and Fig. 5f–j shows a case on 6 September 2010 in which the DR method produced an isolated halocline base depth minimum. In both

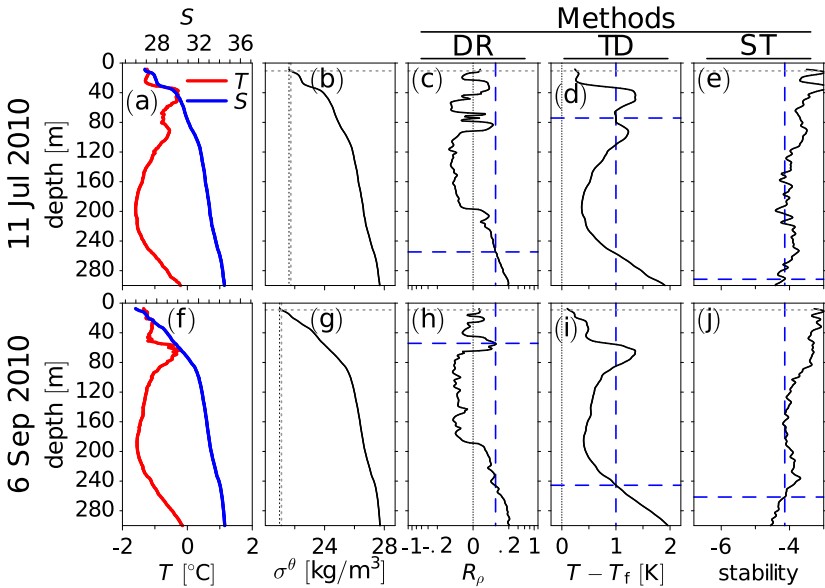

**Figure 5.** As Figure 3, but for two profiles from ITP-33. For 12 July 2010 **(a–e)** the TD method shows an isolated minimum of the halocline base depth, and for 6 September 2010 **(f–j)** the DR method shows an isolated minimum of the halocline base depth. Supplement 6 shows a corresponding figure based on Level I data instead of the more processed Level III data.

cases, the isolated minima are related to a layer of warm PSW around $\sim$80 m (Figs. 5a and f, compare also Fig. 4b). For 11 July 2010, the best estimate of the halocline base depth is provided by the DR method (Fig. 5c). The DR method correctly places the base of the halocline water at a depth, where the salinity gradient changes (Fig. 5a). Stability also decreases markedly at this depth, although the ST method identifies the halocline base about 20 m below this location (Fig. 5e). The TD method places the halocline base at about 80 m in the layer of warm PSW (Fig. 5d), although the salinity gradient below this layer still indicates the presence of a halocline and temperature decreases below this layer, indicating PWW. For 6 September 2010, the DR threshold is exceeded at a local density ratio maximum (Fig. 5h) which is related to a very steep temperature gradient at the base of the PSW (Fig. 5f). While all three methods rely on finding a threshold, the search direction differs. Because the ST method searches upward, the warm PSW layer does not result in isolated depth minima (Figs. 5e and j). Overall, this analysis suggests that the search direction matters for whether spurious halocline base depth minima are detected or not. With the DR and the TD method, we search downward, while with the ST method we search upward. Based on the stability profile in Fig. 5e, searching downward would lead to spurious depth minima in the ST method as well. This implies that the downward search direction in the ST method helps to explain why the ST method yields more robust results with fewer unexpected depth minima appearing in the basin-wise statistics discussed below. For the DR method, on the other hand, Fig. 3a shows that the DR threshold is exceeded also far below the halocline base. Such local DR maxima below the halocline base were also found in the presence of thermohaline staircases (not shown here). Additional DR maxima below the halocline base, such as the one in Fig. 3, prevent us from simply reversing the search direction in the DR algorithm.

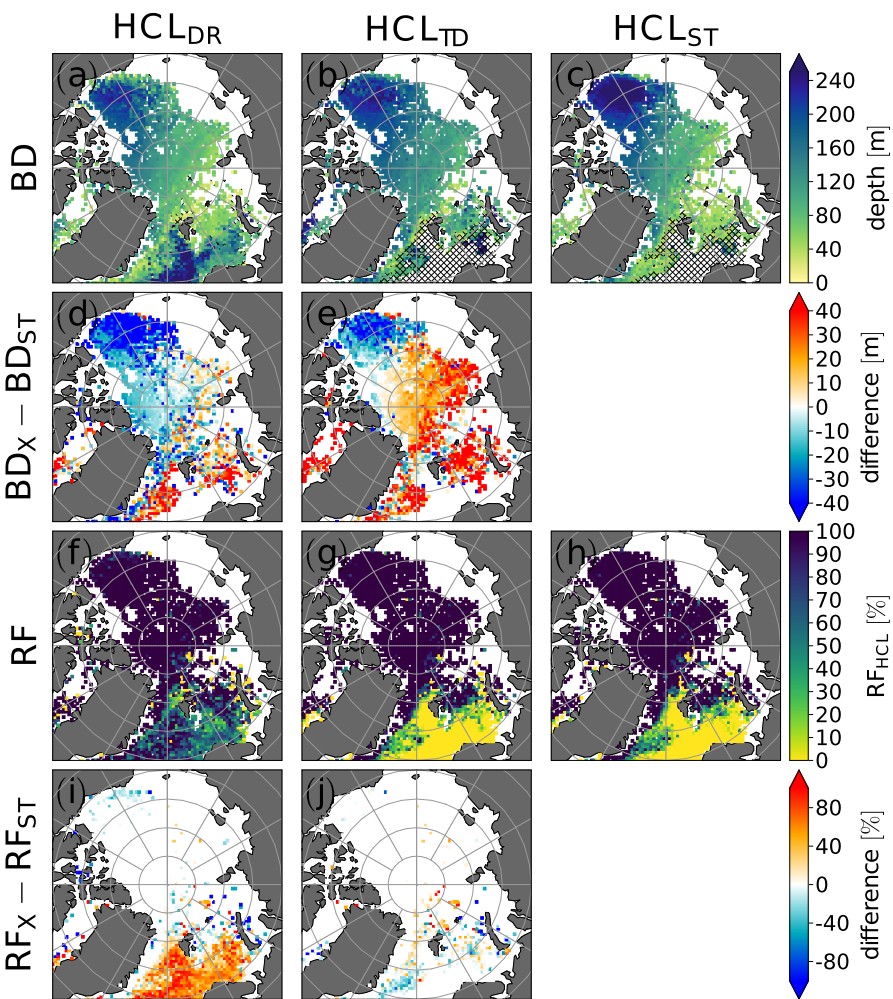

**Figure 6.** Halocline (HCL) base depth (BD) derived by the density-ratio (DR) algorithm **(a)**, the temperature-difference (TD) algorithm **(b)**, and the new stability (ST) algorithm **(c)**. Difference of HCL BD between the DR and the ST algorithm **(d)** and between the TD and the ST algorithm **(e)**. Relative frequency (RF) of HCL occurrence (i.e. number of profiles for which a halocline was detected divided by number of profiles analyzed) **(f–h)** and differences **(i–j)** as in (a–e). In (a–c), points where the relative occurrence frequency of the HCL was below 1% were masked out. Hatching indicates regions where the HCL occurrence frequency is below 25%. Points where the ocean floor is shallower than 100 m were masked out.

## 3.2 Statistical comparison of the halocline base depth and occurrence frequency from different methods

In order to identify differences between the methods used for halocline base detection regarding the geographical distribution of halocline base depth and halocline occurrence frequency (i.e. number of profiles for which a halocline base was detected divided by number of profiles analyzed) we used a simple nearest-neighbor (NN) averaging to produce maps (Fig. 6). In

addition to these maps (which in data-rich regions show a time mean halocline base depth), we computed basin-wise statistics
of halocline base depths. Figure 7 shows relative frequencies of halocline base depth diagnosed with the three methods for the
Eurasian Basin, the Makarov Basin, and the Canada Basin.

As expected, all three methods yield a similar overall spatial distribution of halocline base depth with shallower halocline
base in the Eurasian Basin and the Makarov Basin compared to the Canada Basin (Fig. 6a–c). This spatial pattern is consistent
with Polyakov et al. (2018). In the Eurasian Basin, the time averaged halocline base depth diagnosed with the DR method
(Fig. 6a) and the ST method (Fig. 6c) agree relatively well, while the TD method placed the halocline at greater depth compared
to both other methods (Fig. 6b). For the elliptical area covering the Eurasian Basin in Fig. 7a, the mean base depth was 85.9 m
for the DR method and 92.3 m for the ST method vs. 116.1 m for the TD method. This greater halocline base depth from the
TD method compared to the other two methods is consistent with the previous result for ITP-74. Unlike the DR and the ST
method, which both correctly identified the CHL base during the first months of ITP-74, the TD method placed the halocline
base somewhere in the LHW.

In addition to the moderate difference in the mean base depth between the DR and the ST method, the relative frequency of
halocline base depth in the Eurasian Basin reveals differences between the DR and the ST method, which are not reflected by
the difference of the mean base depths (Fig. 7b). The ST method more often identifies a shallow halocline base ($< 60$ m) in the
Eurasian Basin compared to the DR method, which is consistent with the finding that the ST method apparently captures the
start of new halocline formation from Sect. 3.1. The ST method also detects more frequent cases of halocline base depth below
120 m compared to the DR method (Fig. 7b). The more frequent halocline base depths larger than 120 m in the ST method are
likely related to a deeper halocline base similar to the one found for the ST method for ITP-33 above. Slightly increasing the
stability threshold in the ST method may lead to a better match between the halocline base depth estimate from the ST and
the TD method by moving the halocline base estimated by the ST method upward and by decreasing the sensitivity of the ST
method to new halocline formation. For the elliptical area covering the Makarov Basin in Fig. 7b, the DR method yielded a
mean halocline base depth of 112.3 m, the ST method of 118.1 m, and the TD method again yielded a greater mean halocline
base depth of of 133.5 m.

For the elliptical area covering the Canada Basin, on the other hand, the ST method yielded the largest mean halocline base
depth. The mean halocline base depths for the Canada Basin corresponding to Fig. 7d are 191.4 m for the DR method, 206.6 m
for the TD method, and 219.1 m for the ST method. In the Canada Basin, the ST method detected a halocline base shallower
than 160 m for 0.05% of the profiles, while the DR method detected a halocline base shallower than 160 m for 10.2% of the
profiles and the TD method for 3.5% of the profiles, indicative of isolated base depth minima due to the influence of warm
PSW above PWW (Fig. 7d). Isolated minima very likely also contribute to a more variable (noisier) halocline base depth in
the Canada Basin in the map for the DR method in Fig. 6a and to a lesser extent also in the map for the TD method in Fig. 6b
compared to the ST method (Fig. 6c). The larger average base depth in the Canada Basin in the ST method compared to both
other methods (see also Fig. 6d and e) is, however, not only explained by the isolated depth minima in the DR and the TD
method in Fig. 7d. Instead, more frequent depths greater than 260 m in the ST method compared to the other two methods
(Fig. 7d) contribute to the greater average halocline base depth diagnosed with the ST method, again indicating that a slight

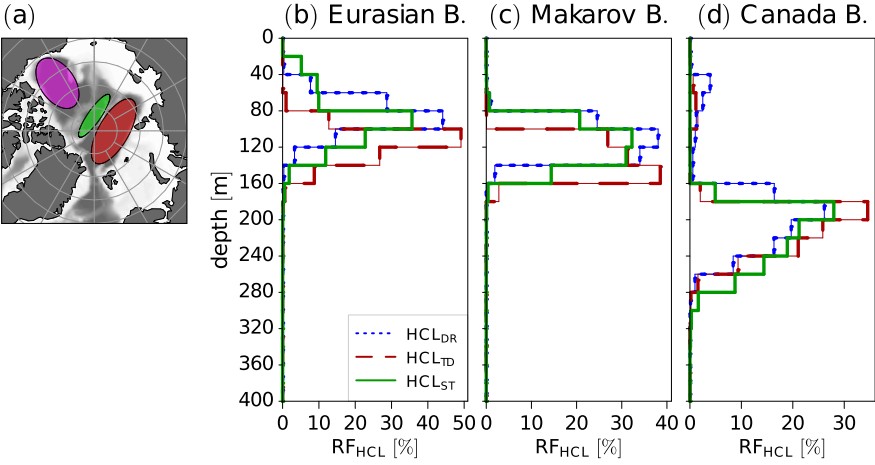

**Figure 7.** Map showing elliptical areas over the Canada Basin (purple), the Makarov Basin (green) and the Eurasian Basin (dark red) and ocean floor depth (grey shading) **(a)**. Relative frequency of halocline (HCL) base depth determined with the density ratio (DR), temperature difference (DR) and the stability (ST) method for the elliptical areas over the Eurasian Basin **(b)**, the Makarov Basin **(c)**, and the Canada Basin **(d)**.

increase of the stability threshold in the ST method would lead to a better agreement between the halocline base depth from the ST and the DR method. While a halocline was almost always detected in the Canada Basin by all three methods, the relative occurrence frequency (defined as the number of profiles in which a halocline base was detected divided by the total number of profiles which were analyzed) varies strongly in the Norwegian Sea (Fig. 6f–h). While the ST method and also the TD method very rarely detected a halocline base in the Norwegian Sea, the DR method frequently detected a halocline base in the Norwegian Sea. Furthermore, the DR method suggests a transition from a deeper halocline in relatively warm water inflow to a shallower halocline further north, which is absent in the other two methods (Fig. 6a–c). When using the DR method for analyzing halocline retreat in these warm water inflow regions, this may lead to different results compared to the two other methods. One reason for the DR method detecting a halocline base at depth could be thermohaline staircases. Overall, the largest differences in halocline detection frequency between the methods (Fig. 6i and j) were found in regions which are prone to sea ice retreat and, according to global climate model results, may also be particularly prone to events, in which large temperature gradients are found directly underneath the SML, and which mainly occur during winter (Metzner et al., 2020). In order to prevent the DR method from identifying a halocline base in relatively warm water, one could either limit the region to which the method is applied (as Polyakov et al., 2018) or else introduce additional constraints on the water temperature. Limiting the region is clearly a sensible choice in a stable climate. However, limiting the region to a region in which a stable (cold) halocline is found at most times limits us in studying regional shifts. With regard to shifts due to climate change, one should be aware that methods differ regarding requirements for halocline base identification and results.

### 3.3 Estimation of cold halostad boundaries

In the Canada Basin, the PWW forms a so-called cold halostad, while the LHW is modified water of Atlantic origin. This leads to a local vertical stability minimum between two local vertical stability maxima (Fig. 5e and j). The local minimum of vertical stability between these two local maxima is associated with a small salinity gradient around the core of the PWW (Fig. 5a and f). Top and base depth timeseries of the CHS derived with the new algorithm described in Sect. 2.2 are shown by dark green lines in Fig. 4. The algorithm was designed to avoid misclassifications of the cold halostad by requiring the difference between the lower stability maximum and the stability minimum (i.e. the depth of the 'stability valley') to be at least $0.2$. Furthermore, the vertical extent of the 'stability valley' was required to be greater than $50\,\mathrm{m}$. This leads to occasional discontinuities in the cold halostad boundary. Figure 4 shows such discontinuities as evidenced by the breaks in the dark green lines. Furthermore, the requirement of a minimum depth leads to shallow halostad layers not being detected.

Collecting all available observations with detected cold halostad boundaries per grid cell leads to the maps of CHS center depth and CHS thickness shown in Fig. 8a and b. The main occurrence region of the cold halostad is the Canada Basin, where Pacific water circulates between the SML and water of Atlantic origin (Shimada et al., 2005). Employing conservative assumptions to avoid a misclassification (including a lower bound of $50\,\mathrm{m}$ for the thickness), we detect a cold halostad layer in the Canada Basin region in Fig. 8b $\sim$70–90% of the time, except in August, September, and October when the mean occurrence frequency ranges from slightly below 60% to slightly below 70% (Fig. 8c). Figures 8a and b show that cold halostad was also detected to the northeast of Greenland, facing Svalbard from the west side of Fram Strait. where glacial cold water may act similar to the Pacific low salinity water (Dmitrenko et al., 2017).

### 4 Summary and discussion

We introduced a new method for determining the halocline base depth based on vertical stability and compared it to the density ratio (DR) and the temperature difference (TD) method. Our main motivation for using a vertical stability (ST) threshold instead of a DR or TD threshold was that vertical stability is more closely related to the role of the halocline as a stable layer which separates the surface mixed layer from warmer Atlantic Water (AW) below and thus acts to protect sea ice. Another objective was to design a particularly robust method with few artifacts. Furthermore, the new ST algorithm does not require masking out pre-defined geographical regions solely based on their location. This is important because in a climate change context, the location of water masses may shift as the Arctic warms or cools. An example is the ongoing atlantification and associated northward shift of ocean conditions previously specific to Nordic Seas/marginal regions (e.g. Athanase et al., 2021). We also devised a new stability-based method to identify the cold halostad (CHS), which is formed by Pacific Winter Water (PWW) in the Canada Basin and also by melt water off the eastern coast of Greenland and Svalbard. To our knowledge, this is the first time that a method for detecting the CHS has been devised and tested.

We found that the DR and the new ST method both correctly identified the base of the cold halocline layer in the Eurasian Basin during the first months of ITP-74, while the TD method placed the halocline base somewhere in the lower halocline water. The analysis of individual profiles after convection in ITP-74 indicated that the new ST method captured the beginning of new

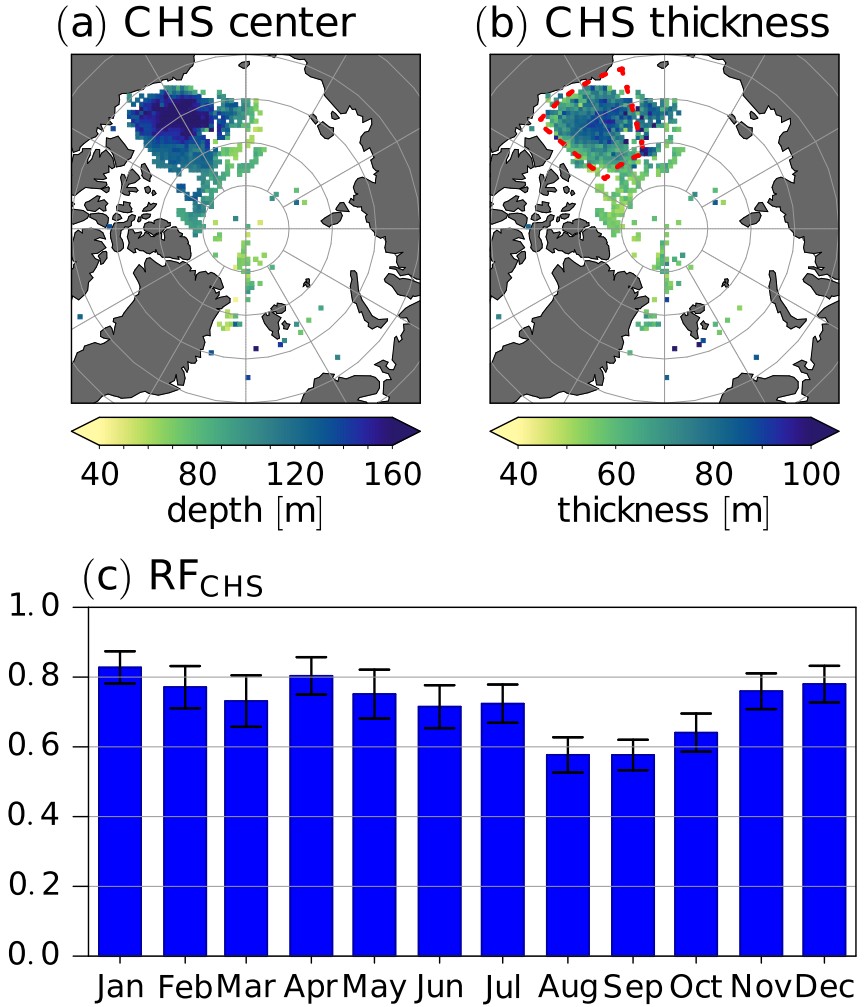

**Figure 8.** Maps of mean CHS center depth **(a)** and CHS thickness **(b)**. Monthly relative occurrence frequency (RF_CHS) of the CHS in the Canada Basin region (area enclosed by red dashed lines in b) and 95% confidence interval for the mean **(c)**.

halocline formation via the convective homogenization and subsequent fresh water capping mechanism proposed by Rudels

et al. (1996). In the Canada Basin, the new method placed the halocline base deeper than the DR method, which correctly identified the halocline base for ITP-33. This disagreement between the DR and the ST method could be reduced by slightly increasing the stability threshold in the ST method. Slightly increasing the stability threshold (which is at present based on an approximate relationship between density ratio and stability) may lead to a better agreement of halocline base depth between the DR and the ST method not only by moving the halocline base from the ST method upward but also by decreasing the

sensitivity of the ST method to new halocline formation. However, instead of simply aiming for a better match between the ST and the DR method, it may ultimately be more desirable to further fine tune the ST threshold based on an empirical analysis of

profiles, similar to how Bourgain and Gascard (2011) determined the DR threshold. On the whole, estimating the ST threshold based on the DR threshold as we did provides fairly reasonable results, even without further fine tuning.

Unlike the two existing methods, the new ST method to detect the halocline base yielded few artificial halocline base depth minima. In the two existing methods, such artifacts were found to be associated with warm Pacific Summer Water on top of cold PWW in the Canada Basin. Because the new method searches for the halocline base from below, such artifacts were avoided, leading to a more robust method, especially compared to the widely used DR method. Unfortunately, because of DR maxima below the cold halocline base (which are for example associated with thermohaline staircases), the search direction in the DR method cannot simply be reversed in order to increase the robustness also of the DR method.

A particularly striking difference between the DR method and the other two methods was found in the Norwegian Sea. While the ST and the TD method almost never detected a halocline in the Norwegian Sea, the DR method frequently detected a halocline base in the Norwegian Sea. Remarkably, the halocline in the DR method decreased north of the Norwegian Sea. This intriguing difference between the methods should be taken into account, for example when studying the effects of warm AW inflow on the cold halocline, especially because warm water inflow regions are particularly prone to react to anthropogenic warming (although the effects of warming on either accelerating or preventing new halocline formation are manifold and changing over time, and destabilization of an existing stable halocline by warming form below is only one potential contributor to increased annual mean net surface heat fluxes from the ocean to the atmosphere in warm water inflow regions found in climate models).

A caveat potentially affecting the halocline base depth determined via the DR and the ST method is that observations show spurious salinity extremes in high-gradient segments which can affect the outcomes. The DR and the ST method may select density values affected by the spurious salinity extremes. Therefore, caution is advised in the application of the DR and the ST method to observations.

Regarding the new method for CHS detection, a case study and an application to a comprehensive dataset yielded encouraging results. The case study suggested that the method correctly identified a layer with a small vertical salinity gradient formed by PWW. This small salinity gradient led to a stability minimum between two local stability maxima which was captured by the new stability method for CHS detection. Because we found it necessary to introduce a constraint on the cold halostad thickness and to set a threshold requirement for the magnitude of the stability minimum, our method suffers from a low detection sensitivity and altogether misses cold halostad layers that are thinner than 50 m. Nevertheless, a cold halostad was frequently detected in the Canada Basin throughout the year and the number of missed detections tended to be small as for ITP-33, albeit with detection rates varying across ITPs tested from almost perfect to a slightly higher mismatch rate (not shown). This suggests that the new stability-based method for CHS detection could be useful for future studies exploring variability and changes of the CHS in the Canada Basin.

One method to advance cold halostad and halocline base detection in the future may lie in the application of artificial intelligence. This would require a-priori manual classification applied to a training and an evaluation dataset. In the absence of objective criteria that work under most circumstances, such manual classification would ultimately have to rely on expert judgment, which may in turn introduce a different set of problems. Rapid advances in AI may help to overcome these problems.

*Code and data availability.* The Ice tethered profiler data (Krishfield et al., 2008; Toole et al., 2011) used in this paper are taken from the website of the Ice-Tethered Profiler program based on Woods Hole Oceanic institution via https://www2.whoi.edu/site/itp/ (last access 23 January 2023). The UDASH-dataset (Behrendt et al., 2018) is available from the PANGAEA data archive at http://doi.org/10.1594/PANGAEA.872931. Sea ice data in Supplement 5 was obtained from https://www.meereisportal.de.

*Author contributions.* E.P.M. devised the new method for determining the Arctic Ocean cold halocline and cold halostad layer depths and performed the data analysis. Both authors contributed equally to writing the manuscript.

*Competing interests.* The authors declare that they have no conflict of interest.

*Acknowledgements.* We thank the two reviewers, I. Polyakov and M. Athanase, for their very insightful and very constructive comments,
especially for their suggestions regarding the analysis of individual profiles, new halocline formation, the structure of the introduction, and basin-wise statistics, which we consider a major contribution to our revised manuscript. The Ice-Tethered Profiler data were collected and made available by the Ice-Tethered Profiler Program based at the Woods Hole Oceanographic Institution (http://www2.whoi.edu/site/itp). Sea ice data in Supplement 5 was obtained from https://www.meereisportal.de (Grosfeld et al., 2016, funded by REKLIM-2013-04). We gratefully acknowledge the funding by the Deutsche Forschungsgemeinschaft (DFG, German Research Foundation) - Projektnummer 268020496
- TRR 172, within the Transregional Collaborative Research Center "ArctiC Amplification: Climate Relevant Atmospheric and SurfaCe Processes, and Feedback Mechanisms (AC)³". We thank J. Chylik for comments that helped to improve the readability of our manuscript.

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
