# Peer review of "Technical note: Determining Arctic Ocean halocline and cold halostad depths based on vertical stability"

_EGUsphere, 2023_

## Author Comment (AC1)

Response to reviewer comments by Igor Polyakov

Thank you very much for your very insightful review and very constructive comments. We repeat your comments (using a dark red font color) below. Our responses are given in black font. Please excuse the delay of our reply. We realize that additional work will be required to prepare a revised manuscript.

The manuscript proposes a new method for estimating the depth of the Cold Halocline Layer (CHL). The topic is important and warrants a lot of attention in the published paper. Thus, I am very positive that the authors have the potential to produce a nice publication.

However, at the current stage, the manuscript suffers from several major shortcomings:

- I did not find a convincing and satisfactory comparison of the three methods defining the halocline. The majority of the materials presented in the manuscript are about the illustration of the application of the methods and not their comparison. This comparison should include an evaluation of each method's performance and an explanation of the benefits of using each method. Right now, the way it is done is not satisfactory. I would like to see, for example, individual temperature (T) and salinity (S) profiles where the authors show what each method provides and explain the physical reasons for that. I found Fig. 3, which is devoted to method comparison, to be hard to read and not informative.

  We have analyzed individual profiles from ITP74 and ITP33. In response to a comment below, we have also re-processed the data at the native vertical resolution. Reprocessing the data at the native vertical resolution affected individual results, but our overall conclusion regarding the robustness of the methods still holds.

  Figure RP1 shows a revised version of Fig. 2. Figure RP2 shows individual profiles before and after winter convection. In Figure RP1, we excluded profiles that started below 15 m. This strongly reduces the effect of noise in determining the SML base without affecting the overall result.

  We find that investigating individual profiles does help to better understand the differences between the methods. Figure RP2 shows profiles

[Figure]

Figure RP1: Modified version of Fig. 1 based on data at the native vertical resolution. Individual profiles at the location of the wedge symbols (∧) below the x-axis in **b** are shown in Fig. RP2. HCL stands for halocline.

before winter convection, during winter convection, and after winter convection from ITP74. Figures RP2a and c for 13 January 2014 show that the TD method misinterprets the halocline (HCL) base. During winter convection (Figs RP2e–h), no HCL was identified. This is because the threshold for identifying the HCL base was already exceeded at the SML base for the DR and the TD method (Fig. RP2f and g), while the threshold was not reached for the ST method (Fig. RP2h). Fig. RP2i for 15 July 2014 (after winter convection) shows freshening and warming near the surface. This indicates that relatively fresh melt or shelf water may have been advected above a colder and saline layer, which had been preconditioned by winter convection. The freshening near the surface leads to a salinity gradient below, and also a stability maximum, which is captured by the ST method (Fig. RP2h). This appears to be consistent with the mechanisms for halocline formation described by Rudels et al. (1996). Rudels et al. (1996) found new halocline formation taking place when relatively fresh shelf waters near the surface were advected above

[Figure]

Figure RP2: Potential temperature $\theta$ and salinity $S$, density ratio, temperature difference, and stability from ITP74 before winter convection on 13 January 2014 (a–d), during winter convection on 20 May 2014 (e–h) and after winter convection on 15 July 2014 (i–l). Vertical dashed lines indicated threshold values. Horizontal dashed lines indicate the halocline (HCL) base determined by the three different methods. Dotted lines indicate the SML base. Dash-dotted lines in (f) and (g) indicate that a threshold for identifying the HCL base was exceeded at the SML base.

denser and saltier water below, limiting winter convection. Rudels et al. (2004) and Alkire et al. (2017) stressed the role of melt water in the warm Atlantic inflow through the Fram Strait and via the Barents Sea for halocline formation via this type of capping mechanism. Fig. RP2h for 15 July 2014 (after winter convection) suggests that the ST method might indeed be useful for identifying the beginning new halocline formation via the Rudels et al. (1996, 2004) mechanism.

Figure RP3 for ITP33, which is a revised version of Fig. 7a (now including

[Figure]

Figure RP3: As Figure RP1 but for ITP33. In addition, the results of the cold halostad (CHS) bound estimation are shown in dark green .

salinity and stability) shows isolated HCL base depth minima for the DR and the TD method, but not for the stability (ST) method.

Figure RP4a–d shows a case from ITP33 on 11 July 2010 in which the TD method produces an isolated HCL base depth minimum and Fig. RP4e–f shows a case on 6 September 2010 in which the DR method produces an isolated HCL base depth minimum. In both cases, the isolated minima are related to the presence of a layer of warm Pacific Summer Water (PSW) around ∼80 m (Figs. RP4a and e). For 11 July 2010, the best estimate of the HCL base depth is provided by the DR method (Fig. RP4b). The DR method correctly places the base of the halocline water at a depth, where the salinity gradient (Fig. RP4a) changes. Stability (Fig. RP4d) also increases markedly at this depth, although the ST method identifies the base of the halocline base about 20 m below this location (Fig. RP4d). The TD method (Fig. RP4c) places the base of the HCL at about 80 m in a layer of warm water, although the salinity gradient below this layer still indicates the presence of a halocline and

[Figure]

Figure RP4: As Figure RP2, but for two profiles from ITP33. For 12 July 2010 (a–d) the TD method shows an isolated minimum of the HCL base depth, and for 6 September 2010 (a–d) the TD method shows an isolated minimum of the HCL base depth.

temperature decreases below this layer, indicating the presence of Pacific Winter Water (PWW). For 6 September 2010, the DR threshold is exceeded at a local maximum which is related to a very steep temperature gradient (Fig. RP4f) at the base of the PSW. While all three methods rely on finding a threshold, the search direction differs. Because the ST method searches upward, the warm PSW layer does not result in isolated depth minima (Figs. RP4d and h).

Overall, the analysis suggests that the search direction matters. With the DR and the TD method, we search downward, while with the ST method we search upward. This may help to explain why the ST method yields more robust results. We would very much like to further emphasize this point in a revised version of our manuscript. We are planning to remove the discussion of the seasonal cycle and instead to focus more on comparing the methods.

- It looks like the authors misinterpret the water structure of the Arctic Ocean, which has direct implications for their comparative analysis of the three methods used for the definition of the halocline base. In the Eurasian Basin, CHL is the layer where T is close to the freezing point (that is why it is called cold) and S rapidly increases. However, below the CHL, there is the second portion of halocline—the lower halocline water—in which T and S increase with depth. The authors should define what they investigate. In the Canadian Basin, this structure is further complicated by the presence of two other halocline water varieties: Pacific summer and winter waters. Showing the Nordic Seas and Siberian shelves should be excluded from their analysis, for example.

We erroneously equated the CHL with halocline (HCL). In a revised version of our introduction, we explain the HCL, CHL, lower halocline waters (LHW), PSW and PWW and switched from CHL to HCL unless we are explicitly referring to the CHL. We are planning to revise the subsequent discussion of the results and the title of our manuscript accordingly. Please refer to our response to the reviewer comments by M. Athanase for the revised version of our introduction.

In Fig. RP5, which represents a revised version of Figs. 5 and 6, we excluded regions shallower than 100 m. Please let us know if you suggest to further adjust this threshold in case we are eventually granted a second round of reviews.

- By the way, it looks to me like the authors miss two golden opportunities to give a beautiful explanation to their methods. a) Fig. 2 from a single ITP record shows a mismatch of results between the authors' stability method (ST) and the density ratio (DR) method after winter convection. I may be wrong, but it is worth checking whether the ST method captures the beginning of the new halocline formation, which is not captured by the two other methods. The halocline formation is a very important topic of Arctic oceanography. b) When the authors analyzed halostad, a possible interpretation may be that they analyzed the boundaries of a variety (winter or summer) of Pacific water. It is worth checking, and if it is true, this may be a nice touch to "sell" the method.

Thank you very much for pointing this out.

[Figure]

Figure RP5: Revised version of Figures 5 and 6 showing HCL base depth and occurrence frequency, excluding plots for SML depths and kriging results.

a) In our response to your first major comment above, we argue that the ST method may indeed capture the beginning of the new halocline formation based on Figs. RP2i and l.

b) Shimada et al. (2005) explained that the cold halostad (CHS) is formed by Pacific Winter Water (PWW). In the revised introduction, we first introduce PWW and PSW and then explain the link between CHS and PWW as follows:

*The PWW could be referred to as a type of cold halocline water (Zhong*

*et al., 2019), although compared to the CHL in the Eurasian Basin, in the PWW, the salinity is lower and the salinity gradient is smaller. This is why Shimada et al. 2005 called the layer that is formed by PWW a cold halostad (CHS). Similarly, interaction between glacial melt water and Arctic water north east of Greenland forms an intermediate low salinity layer with small salinity gradient which is also called a cold halostad (Dmitrenko et al., 2017). Below, we argue that a lower salinity and a smaller salinity gradient in the CHS compared to the LHW below results in two distinct local stability maxima between the base of the LHW and the SML base: The upper stability maximum is associated with an increase of salinity in the upper PWW. The lower stability maximum is associated with another increase of salinity in the LHW. The lower one of these two stability maxima is absent in the presence of a CHL in the Eurasian Basin (except in regions off the eastern coast of Greenland and also Svalbard where melt water also forms a CHS).*

- The way materials presented is often not good enough – see my detailed comments provided to the authors.

  Thank you very much for the suggestions. Please find a revised version of our introduction based on comments by both reviewers in our response to the reviewer comment by M. Athanase. We are planning to revise the rest of the manuscript as well.

Despite these deficiencies, I believe that the authors can improve the manuscript to the level suitable for publications. That is why I give a "major revision" to this manuscript.

We appreciate your patience.

Below are my specific comments.

*Comments:*

1. Line 16: As defined, it is not CHL, but CHL+Lower Halocline Water in the Eurasian Basin.

Yes, thank you very much for pointing this out. The cited values from Polyakov et al. (2018) refer to the base of the halocline (HCL) and not the CHL. We revised the introduction based on comments by both reviewers as described in our response to the reviewer comment by M. Athanase. In the revised version, we introduce HCL, CHL, PWW, PSW, and LHW, and describe the structure of the HCL for the Eurasian Basin and the Amerasian Basin.

2. Line 17: 300m may be correct for the Canadian Basin but not for the Eurasian Basin.

We now write that the Atlantic water is centered near 300 to 500 m in the Eurasian Basin and somewhat deeper in the Canada Basin, citing Macdonald et al. (2015) in addition to Aagaard et al. (1981).

3. Line 43: The halocline has a complex structure indeed, but this was first described long time ago – see papers by e.g. Rudels, Aagaard, Carmack, Steele.

We revised the introduction based on suggestions by both reviewers. We now first explain the structure of the halocline with CHL, PHW, and LHW, in the first paragraph of the introduction. We cite Bertosio et al. (2020) and Bertosio et al. (2022) for their methods used to determine the base of the LHW in the third paragraph. Please refer to our reply to the reviewer comment by M. Athanase for details.

4. Line 57: Same as for line 43, plus add here Pacific waters.

In the revised introduction, we explain the structure of the halcoline in the Amerasian Basin at the end of the first paragraph, mentioning the origins of PSW, PWW, and LHW. We explain that according to Shimada et al. (2005) CHS is formed by PWW.

5. Data description.

RP-9

[Figure]

Figure RP6: Map of observations for each season (starting with MAM for March, April, and May) with blue dots for UDASH profiles and red dots for ITP profiles (a–d). Temporal coverage for UDASH (e) and ITP (f) observations.

- Please provide vertical resolution of original data, time covered by observations, show distribution of data coverage in time and space for annual and seasonal coverage, show separately spatiotemporal coverage provided by ITP and other sources. Provide accuracy of observations.

  Figure RP6 shows the spatiotemporal coverage provided by UDASH and ITP data. The vertical resolution for 45% of the UDASH data is finer than 2 dbar. However, ∼25% of the profiles have a resolution coarser than 20 dbar for the deepest five points. We will further investigate this issue and most likely set a minimum requirement for the vertical resolution before recomputing HCL and CHS depths. The vertical resolution for ITP level III data is 1±0.1 dbar. The accuracy of the sensors used for the ITP observatations is 0.002°C for temperature and 0.002 for salinity (Polyakov et al., 2017). The UDASH dataset (Behrendt et al., 2018) was assembled from a number different sources and platforms.

- I do not understand the need of this complex vertical data interpolation (lines 89-90). I suspect that the original (raw) data have 1 or 2m resolution which should be sufficient for the purposes of the study.

  We re-processed the data without prior vertical interpolation.

- Exclude all points from the shelves where there is no typically halocline found. The same is true for the Nordic Seas.

  We excluded regions shallower than $100\,\text{m}$ (Fig. RP5). Please let us know if you suggest to further adjust this threshold in case we are eventually granted a second round of reviews.

- Regions are shown in Fig 1 but not used. What is the purpose for that?

  Thank you for pointing this out. We are planning to remove the regions from Fig. 1 in a revised version.

1. Line 101: Why T is given in K, not in C?

   This will be changed to °C in a revised version.

2. Line 113: Shallower, not smaller.

   Yes, thank you. This will be changed.

3. Section 2.2.4 and further discussion: All these methods should be illustrated using individual profiles where everything is clearly marked and visible. Same section: You may use definition of the upper Atlantic Water layer by using 0°C isotherm.

Yes, please refer to Figs. RP2 and RP4 above. We replaced the doubled DT-threshold by the 0°C isotherm.

4. Section 2.2.5: The authors may want to clearly define halostad by giving some physical interpretation for this layer.

In the revised version of our introduction, we associate the cold halostad (CHS) with PWW based on Shimada et al. (2005) and explain the origin of the term cold halostad (see above). We also explain that in the presence of a CHS, the underlying LHW accounts for a second stability maximum (compare Fig. RP4a and e)

5. Line 144: Taking 0.001m threshold seems misleading since the original data have a 1m vertical resolution (or even coarser).

We use a vertical interpolation between points, and the 0.001m accounts for rounding errors. We are planning to revise our methods section for clarity.

6. Section 2.3. I found the use of kriging more misleading than helpful in this study.Actually there is no need in that at all since the authors used another much simpler method which serves for the purposes.

We removed the maps with the kriging results (Fig. RP5).

7. Line 168: Filtration of outliers using such a severe threshold of 25/75% (which is less than 1.5 standard deviation) needs an explanation. Sensitivity study where the authors show how sensitive their estimates to different thresholds may be helpful.

Outliers were defined as values outside the interval $[Q1 - 1.5\,IQR, Q3 + 1.5\,IQR]$, where $Q1$ is the first quartile, $Q3$ the third quartile, and $IQR = Q3 - Q1$ is the inter-quartile range. The section on kriging will be omitted.

8. Line 198: An example where the authors misinterpret water masses: I think they found that the CHL disappeared but the lower halocline water (not AW) is below the SML.

We are planning to replace "warm Atlantic water" by "large temperature gradients". The sentence would read: *However, for the model data analyzed in Metzner et al. (2020), large temperature gradients were indeed often found directly underneath the SML during halocline thinning events.*

9. Line 202: Please provide specific for the cases when the criteria were not met: what criteria, why, etc.

Figure 3 will be removed. Instead, we will discuss individual profiles as suggested. The corresponding figures show thresholds used as criteria (Figs. RP2, RP4).

10. Line 204: I did not understand what is written there.

The sentence will be removed. Sometimes, thresholds are exceeded already at the SML base. For these profiles, the HCL base depth is initially computed to be very close to zero (instead of exactly zero because of small rounding errors due to floating point operations). These near-zero values are excluded in the computation of average HCL base depth. When computing the occurrence frequency, these profiles are treated as "no halocline detected". The occurrence frequency is defined as number of profiles in which a cold halocline was detected divided by number of profiles. We will try to explain this better in a revised methods section, focusing more on content and less on technical detail.

11. Fig 2: I think this, plus Fig 7, is a good plot to be used for the method interpretation. I suggest to move line definition from the panel - the authors have space below the map. This case would be nice to illustrate further by using individual profiles of T & S for different regimes.

Thank you very much. Figures 2 and 7 were, of course, inspired by Polyakov et al. (2017). We moved the legend defining the lines and we

followed your suggestion to analyze individual profiles (see above).

Similar plot for the Canadian Basin, plus individual profiles from there, would be a good illustration for regional halocline differences.

Please refer to Figures RP3 and RP4, which we discussed above.

1. 3. Not a good figure. I would completely eliminate it since I found no information in the current version.

   The figure will be eliminated.

2. Section 3.2. What does "occurrence" mean in this context?

   The occurrence frequency is defined as number of profiles in which a cold halocline was detected divided by number of profiles. When a threshold was either exceeded at the SML base or not exceeded at all, the profile was treated as "no halocline detected".

3. Fig 4: Please eliminate point in the Nordic Seas and over the shelves. I suggest to eliminate panels with kriging. The point here is not to show the differences between spatial interpolation methods, but between halocline definition. So, I suggest to show additional panels for differences between methods: e.g. ST-DR and ST-TD. Again, please explain "occurrence". I did not understand the "nearest-neighbor" method: If the nearest data point to the grid point is used, why there was averaging then?

   In the revised version of Fig. 4 (now included in Fig. RP5), we excluded points in regions shallower than $100\,\mathrm{m}$, eliminated the panels with kriging, and included additional panels for differences between methods. Regarding the nearest neighbor method, we first define a grid. Then, all the profiles which are closest to a given grid point are assigned to that grid point. We will explicitly explain this point in the revised methods section.

4. 5: The same as for Fig 4: Please eliminate estimates from kriging and compare methods of halocline definition and not the methods of spatial interpolation. I.e. show CHL-DR minus CHL-ST and CHL-TD minus CHL_ST.

We combined Figs. 4 and 5 in Fig. RP5 and followed your suggestions.

5. Lines 245-247: Often, some analysis of SML is given (like in this paragraph). This is not the topic of the study, first. SML parameters may be given, if they serve the purpose of illustrating the methods. Otherwise, please skip these places. As they are now, they raise questions about newness of these results (e.g. asking for comparison with the previous papers on the subject)

We eliminated the maps showing the SML depths and we are planning to also eliminate the corresponding discussion.

6. Fig 6: The caption is not clear – the difference is not defined. "(e) to (f)" is not clear. Cos fit is not illustrated.

The plot will be eliminated. Difference referred to the difference between seasonal and annual mean.

7. Fig 7. Is good, but a) add T &S (like in Fig 2) b) move line definition from the panel.

Done.

8. Fig 8: What is the point of giving fraction of grid points? What is halostad depth? Give a physical interpretation (Pacific water?).

We reported the fraction of points because the spatial coverage varies with season. CHS depth was defined here as the depth of the center of the CHS. The center was defined as the mean of the upper and the lower bound of the CHS. The upper and the lower bound of the CHS were defined by a threshold stability. This threshold stability was defined as the

mean of the stability minimum in the CHS and the stability maximum in the LHW. We are planning to revise the methods section for clarity.

9. Line 314: What is "semi-saline"?

We replaced the expression "semi-saline" by "low salinity" based on a suggestion in the reviewer comment by A. Athanase.

10. Line 321: How can we see the point made for the Barents Sea?

This discussion will be removed from the revised manuscript. We should have written Nansen Basin.

11. Line 322: This discussion and comparison with Steele work needs much more thourough analysis – not just a single line.

We are planning to revise this discussion based in part on our revised introduction.

12. Line 325 – What does this "lack: mean?

In a revised manuscript, the word "lack" would by replaced by the word "absence".

Thank you very much again and please excuse our delay. We very much appreciate your comments. Should the Editor decide to encourage a re-submission, we will be happy to provide more complete answers to some of your comments and a version containing track changes together with the revised manuscript.

**References**

Macdonald, R. W., Kuzyk, Z. A., and Johannessen, S. C.: It is not just about the ice: a geochemical perspective on the changing Arctic Ocean, J. Environ. Stud. Sci., 5, 288–301, https://doi.org/10.1007/s13412-015-0302-4, 2015.

Zhong, W., Steele, M., Zhang, J., and Cole, S. T.: Circulation of Pacific Winter Water in the Western Arctic Ocean, J. of Geophys. Res. Oceans, 124, 863–881, https://doi.org/10.1029/2018jc014604, 2019.

---

## Author Comment (AC2)

Response to reviewer comments by Marylou Athanase

Thank you very much for your very insightful review and very constructive comments. We repeat your comments (using a dark blue font color) below.

Our responses are given in black font. Please excuse the delay of our reply. We realize that additional work will be required to prepare a revised manuscript.

**Summary:**

The present study proposes a new method for the detection of the cold halocline layer base depth in the Arctic Ocean. The authors define a new criterion based on vertical stability and compare the results to those obtained with two previously used methods: one based on the density ratio, and one based on temperature differences. Using the ITP and UDASH databases, they derive pan-Arctic maps of the cold halocline layer base depth using all three methods.

**General assessment:**

The topic is well within the scope of Ocean Science and is of particular importance given the complex and varying structure of the Arctic halocline throughout basins. However, several crucial points would need to be addressed, regarding the manuscript organization as well as the clarity of the method and completeness of the results. Below are listed my major and minor comments. For these reasons, I recommend that the manuscript undergo major revisions before being potentially suitable for publication.

Major comments:

- The manuscript lacks a clear, physical explanation of what the CHL and cold halostad are, and how the proposed method detects their boundaries. I would also suggest the authors emphasize the goal aimed to be achieved by defining a new criterion, e.g., enabling the robust detection of the CHL base depth across basins and seasons using only one criterion.

Thank you very much for this comment and the more detailed suggestions below. We have drafted a revised version of the introduction based on your comments and on the reviewer comments by I. Polyakov. Regarding the organization of the revised introduction, we followed your detailed suggestions, with

five paragraphs introducing concepts, listing criteria, underlining importance, explaining our motivation, and introducing the organization of the manuscript.

In the revised version (see below), we first introduce layers and water masses. We start with the surface mixed layer and the halocline (HCL). Then, we introduce major components of the HCL: the CHL in the Eurasian Basin, the Pacific Halocline Water (PHW) in the Amerasian Basin, and the Lower Halocline Water (LHW), following the nomenclature of Polyakov et al. (2018). We then introduce Pacific Winter Water (PWW), which forms the cold halostad (CHS), and Pacific Summer Water (PSW).

The more detailed description of the HCL including the introduction of PSW and PWW in the first paragraph of the revised introduction owes to comments and suggestions by I. Polyakov. We strongly agree that introducing these water masses is useful for interpreting the results. For details on this point please refer to our response to the reviewer comments by I. Polyakov (unfortunately, we did not manage to revise the entire manuscript before the end of the discussion phase). We also provide additional detail on CHL formation mechanisms in the first paragraph because I. Polyakov suggested that the ST method may capture the beginning of new CHL formation. We now think that interpreting the results in the light of the Rudels et al. (1996) mechanism supports this idea. Again, for details on this point please refer to our response to the reviewer comments by I. Polyakov.

In the second paragraph of the revised introduction, we underline the importance of the HCL for the present and future of the Arctic Ocean. In the third paragraph, we combined the paragraphs describing the existing criteria to define HCL and CHL base. In the forth paragraph, we describe our original motivation for suggesting the stability (ST) method. Our original motivation was to use a threshold variable to define the HCL base which is more closely related to the role of the HCL as a stable layer which prevents warm Atlantic Water from reaching the SML and reducing sea ice than either the density ratio or the temperature difference. The analysis of individual profiles suggested by both reviewers indicates that the increased robustness of ST method derives mainly from the different search direction in the ST method compared to the two existing methods: in the ST method we search upward instead of downward. This avoids isolated depth minima caused by near-surface features. For

details on this point, again please refer to our response to the reviewer comments by I. Polyakov.

The revised introduction reads as follows (please find a revised introduction including track changes below and a list of references at the end of the document):

[revised manuscript text omitted]

stability maximum that is associated with the CHS and the vertical extent of the CHS. [M.A.]~~In the western Arctic, the CHL splits into an upper CHL and a lower CHL. In between lies water of Pacific origin entering the Arctic Ocean via the Bering Strait. This Pacific water is characterized by lower salinity than Atlantic water, but significantly higher salinity than fresh Arctic surface water (Lin et al., 2021). This leads to an intermediate layer called cold halostad layer Shimada et al., 2005). Similarly, interaction between glacial melt water and Arctic water north east of Greenland forms an intermediate layer of semi-saline water with low salinity gradient which is also called a cold halostad (Dmitrenko et al., 2017).~~ Consistent and robust descriptions of the CHL and cold halostad layer boundaries are important to understand the evolution of the structure of the upper Arctic ocean in the past and the future.

[M.A.]In the next section, we first describe methods to determine the HULK base depth, starting with two existing methods which are used for comparison, i.e. the density ratio (DR) method by Bourgain and Gascard (2011), and the temperature difference (TD) method by Metzner et al. (2020). We then introduce our new stability (ST) method for determining the HCL base depth. In Section 2.4, we propose a new method for estimating the CHS stability maximum depth and CHS extent, which is based on vertical stability as well. In Section 2.5, we describe a method for estimating the SML base depth because the downward search for the DR and the TD threshold starts at the SML base and because the top of the HCL is assumed to coincide with the SML base. In Section 2.6, we introduce observational datasets used for comparison and testing. In Sect. 2, we compare the new ST method for determing the HCL depth to the existing methods and test the new method for determining the CHS depth and extent. The results are summarized in Sect. 3. ~~Here, we propose new diagnostics for the CHL base and the cold halostad layer boundaries and compare our results for the CHL base to the result from two previously suggested methods. The observational datasets on which our analysis is based are introduced in Sect. ??. Details of the density ratio (DR) method by Bourgain and Gascard (2011), the temperature difference (TD) method by Metzner et al. (2020), and the new stability (ST) method are provided in Sect. ??. The top of the CHL is assumed to coincide with the base of the SML (see Sect. ??). We use the kriging method to produce continuous maps of CHL and cold halostad layer boundaries (as explained in Sect. ??). Results are discussed in Sect. ??.~~

- It is my understanding that here, the authors equate the CHL base to the point of maximum stability. Wouldn't the point of maximum stability rather be within the "cline" you are considering, and not necessarily at its base? If I misunderstood, I would suggest the authors clarify their method and the physical reasoning behind it, as stated above.

We are planning to substantially revise our methods section for clarity, and we are planning to take this comment into account. As stated in lines 116f of the original submission, the stability threshold in the ST method was derived based on the density ratio threshold from the DR method. The original motivation behind devising the stability method was that a stable HCL prevents warm Atlantic Water from reaching the surface mixed layer, where it can either melt existing sea ice or prevent sea ice formation. We argue that stability is more closely related to vertical mixing than either the density ratio or the temperature difference.

- The manuscript still lacks an actual evaluation of the performance of each of the detection methods presented here. I like the large scales comparisons, but it lacks some quantitative estimates (which ideally would take into account the varying seasons and basins) and some idea of which method performs best. Because of the diversity of situations, different tests may lead to different rankings of the presented methods. And that is fine, as long as these various results are explicitly presented and discussed.

In order to evaluate the methods and to better understand the reason behind the different robustness, we investigated individual profiles as suggested by both reviewers. The analysis (please refer to our response to the reviewer comments by I. Polyakov for details) suggests that (a) the ST method captured the beginning new halocline formation directly underneath relatively fresh surface water in the Eurasian Basin, (b) the TD method overestimated the HCL depth in the Eurasian Basin, (c) in the Canadian Basin isolated HCL base minima in the DR and the TD method occurred because of a layer of warm Pacific water, and (d) the ST method slightly overestimated the depths of the HCL base in the Canada Basin. These findings are broadly consistent with summary statistics of HCL depth in Fig. RA1. In the Eurasian Basin, the ST method more

RA-11

[Figure]

Figure RA1: Map showing elliptical areas over the Canada Basin (purple), the Makarov Basin (green) and the Eurasian Basin (dark red) and ocean floor depth (grey shading) (a). Relative frequency of HCL base depth determined with the density ratio (DR), temperature difference (DR) and the stability (ST) method for the elliptical areas over the Eurasian Basin (b), the Makarov Basin (c), and the Canada Basin (d).

frequently identifies a HCL base not far below the SML base compared to the DR method. Based on suggestions by I. Polyakov, we now interpret this as an indication of the ST method capturing the beginning of new halocline formation (please refer to our response to the reviewer comments by I. Polyakov for details on this point). The TD method overestimates the CHL depth in the Euarsian Basin (Fig. RA1b). In the Canada Basin (Fig. RA1d), the DR method detects a CHL shallower than 160 m for 2.7% of the profiles and the TD method for 0.5% of the profiles, indicative of isolated maxima due to the influence of near-surface warm Pacific water. Slightly increasing the stability threshold in the ST method may lead to a better match between the TD method by moving the HCL base upward and by decreasing the sensitivity to new HCL formation.

- The organization of the manuscript lacks fluid connections, both in the introduction and in the presentation of the results. Subsections seem to be organized thematically but without a clear logical order. I would suggest the authors consider reorganizing the overall manuscript (introduction and results) and the abstract as follows:

1) Clear introduction of what SML, CHL, and cold halostad are, and why they matter (as already done in part for the CHL).

2) Presenting the previous methods that have been used to define these layers, and underlining eventual (knowledge) gaps in these methods.

3) Introducing the new method, the physical reasoning behind its development, and the goal it aims to achieve.

4) Demonstrating how the results of the new method compare to results from previous methods (qualitatively and quantitatively, by adding some basin-wise statistics for example), and what are the gains of the new method.

We have revised the introduction, computed basin-wise statistics (see above), and now use individual profiles in our comparison of the results (for the latter, please refer to our response to the reviewer comments by I. Polyakov). Unfortunately, we have not yet managed to revise the entire manuscript. We would very much appreciate if the Editor provides us an opportunity to revise the rest of the manuscript as well.

Minor comments:

In the introduction in general:

It would be good to improve logical connections between paragraphs. For example, L26-27: what is the connection between Atlantification discussions and CHL characteristics? You could finish the previous paragraph by commenting on possible changes in the strength of stratification within the CHL that could either boost or hinder further Atlantification.

We agree that it would be good to improve the logical connections between paragraphs. However, the first paragraph of the revised introduction is entirely

devoted to introducing the different layers and water masses. Atlantification is now only mentioned in the second paragraph, which underlines the importance of these concepts. This provides a very nice and clear separation, but makes a smooth transition difficult.

In general, I would suggest reshaping your introduction as follows (as stated above):

- Defining the broad concepts of cold halocline layer and cold halostad

We followed this suggestion, but also introduce water masses such as PSW and PWW and discuss CHL formation mechanisms, especially the Rudels et al. (1996) mechanism, as explained in our response to your first major comment.

- Underlining their importance for our understanding of the present and future Arctic ocean characteristics

We now introduce layers and water masses in the first paragraph and comment on their importance in the second paragraph.

- Listing criteria for their definition used so far, and eventually pros and cons

We list the criteria. We found a comparison of HCL base depths computed with different methods in Bertosio et al. (2022). But as far as we know, there has been no systematic comparison of the pros and cons so far. We think that the devised methods are all very useful for the purpose they were designed for.

- Explaining clearly the motivation behind creating a new set of criteria, and the problem you aim to address

Regarding the HCL base, the existing thresholds are not as closely related to the role of the HCL as a stable layer which prevents mixing (although Section 2.2.4 of the original manuscript shows that density ratio and stability are indeed related). But we consider this a feature of the methods, and not necessarily a problem. Each of the methods targets different aspects, which may be more or less relevant to the question at hand.

- Finally, introduce the organization of the manuscript (as you already do)

RA-14

We revised the paragraph, and we are eager to also revise the rest of the manuscript. (Once again, sorry for the delay. We did not manage to finish revising the manuscript before the end of the discussion phase).

L22–26: Remind the readers in 1-2 sentences what Atlantification is. In fact, you partly do so in the following sentence, but this should come earlier. And isn't ice loss a symptom/characteristic of Atlantification, too?

The revised sentence reads as follows: *Retreating sea ice, increased surface heat flux and the retreat of the CHL have been called atlantification of the Eurasian Basin (Polyakov et al., 2017).*

L38–39: Here, and generally throughout the manuscript, please define concepts as early as they appear. This sentence should come as you mention the density ratio, 2 lines above. In short, put the sentence line 37-38 at the end of this paragraph.

Thank you very much. We moved the sentence in lines 37-38 to the end of the paragraph. In the overview of halocline layers and waters in the first paragraph of the introduction, we provide short definitions of the Pacific Halocline Water (PHW) and the Lower Halocline Water (LHW) in parenthesis before expanding on details further below in the same paragraph.

L34–52: It would be best to reshape this series of 3 paragraphs into one, listing all existing used criteria to define the base of the CHL and eventually their pro and cons.

We combined the three paragraphs. Please refer to our response above regarding a discussion of the pros and cons already in the introduction. Thanks to the reviewers' suggestion to focus on individual profiles, we consider ourselves in a much better position to provide a more substantial discussion of the pros and cons in the results and the conclusion section of a revised manuscript.

L57: You already introduced this concept above when citing Bertosio (2020) and (2022). Would be best to merge the descriptions of upper and lower CHL into one paragraph here, keeping the relevant citations.

We merged the paragraphs and kept the citations. The relevant concepts are introduced in the first paragraph of the revised introduction.

L75: "Level III data": This processing level naming convention is rather opaque for unfamiliar readers. State more clearly what this entails (visual inspection, vertical interpolation, salinity spikes or bias corrections... etc)

The data processing for the ITP data is described by Krishfield et al. `http://www.whoi.edu/fileserver.do?id=35803&pt=2&p=41486`. Producing Level III data included removal of corrupted data, corrections for the sensor response behavior, calibrations, and final screening of spurious outliers. We will mention this in a revised version of our methods section.

L91: "Gaussian filter": as above, please introduce each processing method explicitly.

We are planning to revise the sentence as follows: *In order to reduce noise, the data was smoothed using a standard one-dimensional Gaussian filter (convolution with a Gaussian function, e.g. Deng and Cahill, 1993) with a standard deviation of 2 dbar and a truncation at ±10 dbar.*

Where the standard deviation and the truncation are in dbar instead of m because we have re-processed the data at the original vertical resolution without prior re-gridding and reduced the filter width and truncation.

Please note that Gaussian filters are a standard tool in signal processing (`https://en.wikipedia.org/wiki/Gaussian_filter`). Gaussian filters are generally more efficient at filtering high frequency noise compared to box filters (also known as running mean). Bourgain and Gascard (2001) used box filters of different widths for different variables.

L89: Is this reasonable for all profiles? Could some profiles, especially the oldest ones, have a vertical sampling of 5 to 10 m? If that is the case, please state so and briefly discuss why you think such a high-resolution interpolation is appropriate and reasonable. I would also suggest the authors consider a vertical resolution that is less fine and closer to the native profiles' vertical resolutions.

We reprocessed the data at the native resolution.

L139: it seems this sentence has grammar issues.

Yes, thank you. The sentence should have read:

RA-16

*A cold halostad will only be recognized by the algorithm if the vertical distance between the deeper stability maximum and the first upper occurrence of that same stability value is at least 50 m, and this stability layer has at least a relative depth of 0.2 in $log_{10}(N^2)$.*

This section will be revised for clarity.

L141:..."only the lowest of these layers is identified as a cold halostad." Why is that, physically?

The sentence in the original manuscript stated that in *rare cases, in which more than one layer fulfills these conditions, only the lowest of these layers is identified as a cold halostad*, although we had not encountered any such cases, and only now added a few lines of code to identify and count such cases. It turned out that no such case occurred. We are planning to revise the statement accordingly.

L169: 25th/75th percentile: This is an extremely stringent test. Can you explain why you took such a high threshold? As the other reviewer stated, it would be good to know how sensitive your results are to the "outlier" threshold you chose.

Outliers were defined as values outside the interval $[Q1 - 1.5\,IQR, Q3 + 1.5\,IQR]$, where $Q1$ is the first quartile, $Q3$ the third quartile, and $IQR = Q3 - Q1$ is the inter-quartile range. The section on kriging will be omitted.

L254: In general, I would try to limit references to previous specific figures from previous papers. It is easier if you directly remind the readers what were the findings shown that figure through text, citing the source paper, in discussing your results in light of it.

This discussion will be removed. We have provided updated figures in which we removed the panels showing maps of the SML in our response to the reviewer comments by I. Polyakov.

L314: If you mean "comparatively low-salinity" then use "low salinity".

Yes, thank you very much. We followed your suggestion and corrected this expression in the revised introduction.

RA-17

Fig. 2 and 7: I like these figures, and it would be great to have one of such plots for -when possible- each subregion and season. Even on one ITP, you could for example indicate when the buoys are in Canada vs. Makarov vs. Amundsen / Nansen basins.

Thank you very much for liking these figures. They were inspired by Polyakov et al. (2017). Our response to the reviewer comments by I. Polyakov includes revised versions of Figs. 2 and 7. Fig. RA1 shows statistics for selected basins.

Fig. 3: This is an interesting visualization, but it is rather under-used in the manuscript. I would suggest the author consider replacing it with plots presenting the vertical profiles, either in the introduction to present your various concepts (SML, upper and lower CHL, cold halostad), or in your results by grouping profiles in similar regions or seasons.

We analyzed individual profiles (please refer to our response to the reviewer comments by I. Polyakov). Fig. 3 will be replaced. Regarding summary statistics, we are planning to include Fig. RA1 in a revised version of our manuscript.

Fig. 4 and 5: I still do not quite get the goal of using 2 interpolation methods (NN and kriging). If there is no other objective that showing the pan-Arctic results in 2 different ways, I would suggest the authors pick one of these methods and eliminate the other, in order to make the manuscript more fluid.

The kriging will be eliminated.

Thank you very much again and please excuse our delay. We very much appreciate your comments. Should the Editor decide to encourage a re-submission, we will be happy to provide more complete answers to some of your comments and a version containing track changes together with the revised manuscript.

**References**

[revised manuscript text omitted]

---

## Author Response (AR1)

Dear Ilker,

Thank you very much for handling our manuscript. Our "final author reply to the editor" is structured as follows:

Best regards,
Enrico and Marc

**Updated response to reviewer comments by Igor Polyakov**

Thank you very much for your very insightful review and very constructive comments. We repeat your comments (using a dark red font color) below. Our responses are given in black font. Please find a track changes version at the end of this document.

The manuscript proposes a new method for estimating the depth of the Cold Halocline Layer (CHL). The topic is important and warrants a lot of attention in the published paper. Thus, I am very positive that the authors have the potential to produce a nice publication.

However, at the current stage, the manuscript suffers from several major shortcomings:

- I did not find a convincing and satisfactory comparison of the three methods defining the halocline. The majority of the materials presented in the manuscript are about the illustration of the application of the methods and not their comparison. This comparison should include an evaluation of each method's performance and an explanation of the benefits of using each method. Right now, the way it is done is not satisfactory. I would like to see, for example, individual temperature (T) and salinity (S) profiles where the authors show what each method provides and explain the physical reasons for that. I found Fig. 3, which is devoted to method comparison, to be hard to read and not informative.

We have analyzed individual profiles from ITP-74 and ITP-33. In response to a comment below, we have also re-processed the data at the native vertical resolution. Reprocessing the data at the native vertical resolution affected individual results (see e.g. Figs. 2 and 4 of the revised manuscript), but our overall conclusion regarding the robustness of the methods still holds. In Figure 2, we excluded profiles that started below 15 m. This strongly reduces the effect of noise in determining the SML base without affecting the overall result.

We find that investigating individual profiles does help to better understand the differences between the methods. Figure 3 shows profiles before winter convection, during winter convection, and after winter convection from ITP74. Figures 3a and c for 13 January 2014 show that the TD method misinterprets the halocline base. During winter convection (Figs 3e–h), no halocline was identified. This is because the threshold for identifying the halocline base was already exceeded at the SML base for the DR and the TD method (Fig. 3f and g), while the threshold was not reached for the ST method (Fig. 3h). Fig. 3i for 15 July 2014 (after winter convection) shows freshening and warming near the surface. This indicates that relatively fresh melt or shelf water may have been advected above a colder and saline layer, which had been preconditioned by winter convection. The freshening near the surface leads to a salinity gradient below, and also a stability maximum, which is captured by the ST method (Fig. 3h). This appears to be consistent with the mechanisms for halocline formation described by Rudels et al. (1996). Rudels et al. (1996) found new halocline formation taking place when relatively fresh shelf waters near the surface were advected above denser and saltier water below, limiting winter convection. Rudels et al. (2004) and Alkire et al. (2017) stressed the role of melt water in the warm Atlantic inflow through the Fram Strait and via the Barents Sea for halocline formation via this type of capping mechanism. Fig. 3h for 15 July 2014 (after winter convection) suggests that the ST method might indeed be useful for identifying the beginning new halocline formation via the Rudels et al. (1996, 2004) mechanism.

Figure 4 for ITP-33, which is a revised version of Fig. 7a (now including salinity and stability) shows isolated halocline base depth minima for the DR and the TD method, but not for the stability (ST) method.

Figure 5a–d shows a case from ITP-33 on 11 July 2010 in which the TD method produces an isolated halocline base depth minimum and Fig. 5e–f shows a case on 6 September 2010 in which the DR method produces an isolated halocline base depth minimum. In both cases, the isolated minima are related to the presence of a layer of warm Pacific Summer Water (PSW) around ∼80 m (Figs. 5a and e). For 11 July 2010, the best estimate of the halocline base depth is provided by the DR method (Fig. 5b). The DR method correctly places the base of the halocline water at a depth, where the salinity gradient (Fig. 5a) changes. Stability (Fig. 5d) also decreases markedly at this depth, although the ST method identifies the base of the halocline base about 20 m below this location (Fig. 5d). The TD method (Fig. 5c) places the base of the halocline at about 80 m in a layer of warm water, although the salinity gradient below this layer still indicates the presence of a halocline and temperature decreases below this layer, indicating the presence of Pacific Winter Water (PWW). For 6 September 2010, the DR threshold is exceeded at a local maximum which is related to a very steep temperature gradient (Fig. 5f) at the base of the PSW. While all three methods rely on finding a threshold, the search direction differs. Because the ST method searches upward, the warm PSW layer does not result in isolated depth minima (Figs. 5d and h).

Overall, the analysis suggests that the search direction matters. With the DR and the TD method, we search downward, while with the ST method we search upward. This may help to explain why the ST method yields more robust results. We have emphasized this point in the revised version of our manuscript. The discussion of the seasonal cycle was removed. The revised manuscript focusses more on comparing the methods. Please refer to the track changes version for details and further information.

- It looks like the authors misinterpret the water structure of the Arctic Ocean, which has direct implications for their comparative analysis of the three methods used for the definition of the halocline base. In the Eurasian Basin, CHL is the layer where T is close to the freezing point (that is why it is called cold) and S rapidly increases. However, below the CHL, there is the second portion of halocline—the lower halocline water—in which T and S increase with depth. The authors should define

what they investigate. In the Canadian Basin, this structure is further complicated by the presence of two other halocline water varieties: Pacific summer and winter waters. Showing the Nordic Seas and Siberian shelves should be excluded from their analysis, for example.

We erroneously equated the CHL with halocline. In the revised version of our introduction, we explain the halocline, CHL, lower halocline waters (LHW), PSW and PWW and switched from CHL to halocline unless we are explicitly referring to the CHL. We revised the subsequent discussion of the results and the title of our manuscript accordingly. Please refer to the revised version of our manuscript for details.

In Fig. 6, which represents a revised version of Figs. 5 and 6, we excluded regions shallower than 100 m. Please let us know if you suggest to further adjust this threshold.

- By the way, it looks to me like the authors miss two golden opportunities to give a beautiful explanation to their methods. a) Fig. 2 from a single ITP record shows a mismatch of results between the authors' stability method (ST) and the density ratio (DR) method after winter convection. I may be wrong, but it is worth checking whether the ST method captures the beginning of the new halocline formation, which is not captured by the two other methods. The halocline formation is a very important topic of Arctic oceanography. b) When the authors analyzed halostad, a possible interpretation may be that they analyzed the boundaries of a variety (winter or summer) of Pacific water. It is worth checking, and if it is true, this may be a nice touch to "sell" the method.

Thank you very much for pointing this out.

a) In our response to your first major comment above, we argue that the ST method may indeed capture the beginning of the new halocline formation based on Figs. 3i and l. We added this point to the abstract of the revised version. Your reviewer comment is cited at the end of the introduction.

b) Shimada et al. (2005) explained that the cold halostad (CHS) is formed by Pacific Winter Water (PWW). In the revised introduction, we first introduce PWW and PSW and then explain the link between CHS

and PWW. Please refer to the revised version for details.

- The way materials presented is often not good enough – see my detailed comments provided to the authors.

  Thank you very much for the suggestions. We revised the entire manuscript. Please refer to the track changes version at the end of this document.

Despite these deficiencies, I believe that the authors can improve the manuscript to the level suitable for publications. That is why I give a "major revision" to this manuscript.

We appreciate your patience.

Below are my specific comments.

*Comments:*

1. Line 16: As defined, it is not CHL, but CHL+Lower Halocline Water in the Eurasian Basin.

   Yes, thank you very much for pointing this out. The cited values from Polyakov et al. (2018) refer to the base of the halocline and not the CHL. We revised the introduction based on comments by both reviewers. In the revised version, we introduce halocline, CHL, PWW, PSW, and LHW, and describe the structure of the halocline for the Eurasian Basin and the Amerasian Basin. Please refer either to the track changes version or to the revised mansucript for details.

2. Line 17: 300m may be correct for the Canadian Basin but not for the Eurasian Basin.

   We now write that the Atlantic water is centered near 300 to 500 m in the Eurasian Basin and somewhat deeper in the Canada Basin, citing Macdonald et al. (2015) in addition to Aagaard et al. (1981).

3. Line 43: The halocline has a complex structure indeed, but this was first described long time ago – see papers by e.g. Rudels, Aagaard, Carmack, Steele.

We revised the introduction based on suggestions by both reviewers. We now first explain the structure of the halocline with CHL, PHW, and LHW, in the first paragraph of the introduction. We cite Bertosio et al. (2020) and Bertosio et al. (2022) for their methods used to determine the base of the LHW in the third paragraph.

4. Line 57: Same as for line 43, plus add here Pacific waters.

In the revised introduction, we explain the structure of the halcoline in the Amerasian Basin at the end of the first paragraph, mentioning the origins of PSW, PWW, and LHW. We explain that according to Shimada et al. (2005) CHS is formed by PWW.

5. Data description.

- Please provide vertical resolution of original data, time covered by observations, show distribution of data coverage in time and space for annual and seasonal coverage, show separately spatiotemporal coverage provided by ITP and other sources. Provide accuracy of observations.

  Figure 1 of the revised shows the spatiotemporal coverage provided by UDASH and ITP data. UDASH contains low resolution profiles. In the original version of our manuscrpit, we had erroneously taken even profiles into account, in which the resolution is clearly insufficient. As explained in the revised version, we now filter the data. In the revised version, only profiles with a vertical resolution finer than 2.5 dbar in the upper 300 m and a vertical resolution finer than 5 dbar elsewhere were used. The vertical resolution for ITP level III data is $1\pm0.1$ dbar. The accuracy of the sensors used for the ITP observatations is 0.002°C for temperature and 0.002 for salinity (Polyakov et al., 2017). This is now stated in the revised manuscript.

The UDASH dataset (Behrendt et al., 2018) was assembled from a number different sources and platforms.

- I do not understand the need of this complex vertical data interpolation (lines 89-90). I suspect that the original (raw) data have 1 or 2m resolution which should be sufficient for the purposes of the study.

  We re-processed the data without prior vertical interpolation.

- Exclude all points from the shelves where there is no typically halocline found. The same is true for the Nordic Seas.

  We excluded regions shallower than $100\,\mathrm{m}$ (Fig. **??**). Please let us know if you suggest to further adjust this threshold in case we are eventually granted a second round of reviews.

- Regions are shown in Fig 1 but not used. What is the purpose for that?

  Thank you for pointing this out. We removed the regions from Fig. 1 in the revised version.

1. Line 101: Why T is given in K, not in C?

   This was changed to °C in the revised version.

2. Line 113: Shallower, not smaller.

   Yes, thank you. This was changed.

3. Section 2.2.4 and further discussion: All these methods should be illustrated using individual profiles where everything is clearly marked and

visible. Same section: You may use definition of the upper Atlantic Water layer by using 0°C isotherm.

Yes, please refer to Figs. 3 and 5 of the revised manuscript. We replaced the doubled DT-threshold by the 0°C isotherm.

4. Section 2.2.5: The authors may want to clearly define halostad by giving some physical interpretation for this layer.

In the revised version of our introduction, we associate the cold halostad (CHS) with PWW based on Shimada et al. (2005) and explain the origin of the term cold halostad (see above). We also explain that in the presence of a CHS, the underlying LHW accounts for a second stability maximum (compare Fig. 5a and e)

5. Line 144: Taking 0.001m threshold seems misleading since the original data have a 1m vertical resolution (or even coarser).

We use a vertical interpolation between points, and the 0.001m accounts for rounding errors. We revised our methods section for clarity.

6. Section 2.3. I found the use of kriging more misleading than helpful in this study.Actually there is no need in that at all since the authors used another much simpler method which serves for the purposes.

We removed the maps with the kriging results (see Fig. 6 of the revised manuscript).

7. Line 168: Filtration of outliers using such a severe threshold of 25/75% (which is less than 1.5 standard deviation) needs an explanation. Sensitivity study where the authors show how sensitive their estimates to different thresholds may be helpful.

Outliers were defined as values outside the interval $[Q1 - 1.5\,IQR, Q3 + 1.5\,IQR]$, where $Q1$ is the first quartile, $Q3$ the third quartile, and $IQR = Q3 - Q1$ is the inter-quartile range. The section on kriging was omitted.

8. Line 198: An example where the authors misinterpret water masses: I think they found that the CHL disappeared but the lower halocline water (not AW) is below the SML.

We omitted the sentence containing the reference to "warm Atlantic water" from the revised mansucript. In a sentence in Section 3.2 of the revised manuscript, we now refer to "large temperature gradients" found directly underneath the SML.

9. Line 202: Please provide specific for the cases when the criteria were not met: what criteria, why, etc.

Figure 3 was removed. Instead, we discuss individual profiles as suggested. The corresponding figures show thresholds used as criteria (Figs. 3, 5).

10. Line 204: I did not understand what is written there.

The sentence was removed. Sometimes, thresholds are exceeded already at the SML base. For these profiles, the halocline base depth is initially computed to be very close to zero (instead of exactly zero because of small rounding errors due to floating point operations). These near-zero values are excluded in the computation of average halocline base depth. When computing the occurrence frequency, these profiles are treated as "no halocline detected". The occurrence frequency is defined as number of profiles in which a cold halocline was detected divided by number of profiles. We tried to explain this better in the methods section of the revised manuscript. We re-wrote the methods section to focus more on content and less on technical detail. Please refer to the track changes version for details.

11. Fig 2: I think this, plus Fig 7, is a good plot to be used for the method interpretation. I suggest to move line definition from the panel - the authors have space below the map. This case would be nice to illustrate further by using individual profiles of T & S for different regimes.

Thank you very much. Figures 2 and 7 were, of course, inspired by Polyakov et al. (2017). We moved the legend defining the lines and we followed your suggestion to analyze individual profiles (see above).

Similar plot for the Canadian Basin, plus individual profiles from there, would be a good illustration for regional halocline differences.

Please refer to Figures 4 and 5 of the revised manuscript and the corresponding discussion.

1. 3. Not a good figure. I would completely eliminate it since I found no information in the current version.

The figure was eliminated.

2. Section 3.2. What does "occurrence" mean in this context?

The occurrence frequency is defined as number of profiles in which a cold halocline was detected divided by number of profiles analyzed. When a threshold was either exceeded at the SML base or not exceeded at all, the profile was treated as "no halocline detected". In the revised manuscript, we introduce the term occurence frequency where it first occurs in Sect. 3.2.

3. Fig 4: Please eliminate point in the Nordic Seas and over the shelves. I suggest to eliminate panels with kriging. The point here is not to show the differences between spatial interpolation methods, but between halocline definition. So, I suggest to show additional panels for differences between methods: e.g. ST-DR and ST-TD. Again, please explain "occurrence". I did not understand the "nearest-neighbor" method: If the nearest data point to the grid point is used, why there was averaging then?

In the revised version of Fig. 4 (now included in Fig. 6), we excluded points in regions shallower than 100 m, eliminated the panels with kriging, and included additional panels for differences between methods. Regarding the nearest neighbor method, we first define a grid. Then, all

the profiles which are closest to a given grid point are assigned to that grid point. We will explicitly explain this point in the revised methods section.

4. 5: The same as for Fig 4: Please eliminate estimates from kriging and compare methods of halocline definition and not the methods of spatial interpolation. I.e. show CHL-DR minus CHL-ST and CHL-TD minus CHL_ST.

    We combined Figs. 4 and 5 of the original manuscript in Fig. 6 of the revised manuscript and followed your suggestions.

5. Lines 245-247: Often, some analysis of SML is given (like in this paragraph). This is not the topic of the study, first. SML parameters may be given, if they serve the purpose of illustrating the methods. Otherwise, please skip these places. As they are now, they raise questions about newness of these results (e.g. asking for comparison with the previous papers on the subject)

    We eliminated the maps showing the SML depths and the corresponding discussion.

6. Fig 6: The caption is not clear – the difference is not defined. "(e) to (f)" is not clear. Cos fit is not illustrated.

    The plot was eliminated. Difference referred to the difference between seasonal and annual mean.

7. Fig 7. Is good, but a) add T &S (like in Fig 2) b) move line definition from the panel.

    Done.

8. Fig 8: What is the point of giving fraction of grid points? What is halostad depth? Give a physical interpretation (Pacific water?).

We reported the fraction of points because the spatial coverage varies with season, and we now also refer to this information in the discussion of the figure. CHS depth was defined here as the depth of the center of the CHS. The center was defined as the mean of the upper and the lower bound of the CHS. The upper and the lower bound of the CHS were defined by a threshold stability. This threshold stability was defined as the mean of the stability minimum in the CHS and the stability maximum in the LHW. We revised the methods section for clarity.

9. Line 314: What is "semi-saline"?

   We replaced the expression "semi-saline" by "low salinity" based on a suggestion in the reviewer comment by A. Athanase.

10. Line 321: How can we see the point made for the Barents Sea?

    This discussion will was removed from the revised manuscript. We should have written Nansen Basin.

11. Line 322: This discussion and comparison with Steele work needs much more thourough analysis – not just a single line.

    We now discuss new halocline formation. We omitted the reference to Steele 1995. Instead, we simply point out that the methods produce qualitatively different results, with halocline base depth decreasing north of the Norwegian Sea for the TD method.

12. Line 325 – What does this "lack: mean?

    The sentence was removed from the revised manuscript. We should have written absence instead of lack.

Thank you very much again. We very much appreciate your comments. Please refer to the track changes version for a detailed description of all the revisions. Because of the substantial revisions in almost all sections, this response is still limited to summarizing and outlining major changes. We have re-written

parts of the methods section for clarity, revised the results section based on the very constructive comments and suggestions by both reviewers, and also adapted the title, the abstract and the summary section accordingly. We did not attempt to describe and motivate each of the individual changes in this response. Instead, we marked the changes in the track changes version. In some cases, including cases in which we noted and removed errors, we also motivated and explained changes in the track changes version. In a few places, we changed the draft version of the revised introduction that was included in the original response to the reviewer comment by M. Athanase. However, we completely kept track only of the changes compared to the original manuscript and not to this draft. We very much appreciate your effort in reviewing this technical note and consider the suggestions from the reviews a key contribution to the revised manuscript.

**Updated response to reviewer comments by Mary-lou Athanase**

Thank you very much for your very insightful review and very constructive comments. We repeat your comments (using a dark blue font color) below.

Our responses are given in black font. Please find a track changes version at the end of this document.

**Summary:**

The present study proposes a new method for the detection of the cold halocline layer base depth in the Arctic Ocean. The authors define a new criterion based on vertical stability and compare the results to those obtained with two previously used methods: one based on the density ratio, and one based on temperature differences. Using the ITP and UDASH databases, they derive pan-Arctic maps of the cold halocline layer base depth using all three methods.

**General assessment:**

The topic is well within the scope of Ocean Science and is of particular importance given the complex and varying structure of the Arctic halocline throughout basins. However, several crucial points would need to be addressed, regarding the manuscript organization as well as the clarity of the method and completeness of the results. Below are listed my major and minor comments. For these reasons, I recommend that the manuscript undergo major revisions before being potentially suitable for publication.

Major comments:

- The manuscript lacks a clear, physical explanation of what the CHL and cold halostad are, and how the proposed method detects their boundaries. I would also suggest the authors emphasize the goal aimed to be achieved by defining a new criterion, e.g., enabling the robust detection of the CHL base depth across basins and seasons using only one criterion.

Thank you very much for this comment and the more detailed suggestions below. We have revised the introduction based on your comments and on the reviewer comments by I. Polyakov (please refer to the track changes version for

details). Regarding the organization of the revised introduction, we followed your detailed suggestions, with five paragraphs introducing concepts, listing criteria, underlining importance, explaining our motivation, and introducing the organization of the manuscript.

In the revised version of the introduction, we first introduce layers and water masses. We start with the surface mixed layer and the halocline. Then, we introduce major components of the halocline: the CHL in the Eurasian Basin, the Pacific Halocline Water (PHW) in the Amerasian Basin, and the Lower Halocline Water (LHW), following the nomenclature of Polyakov et al. (2018). We then introduce Pacific Winter Water (PWW), which forms the cold halostad (CHS), and Pacific Summer Water (PSW).

The more detailed description of the halocline including the introduction of PSW and PWW in the first paragraph of the revised introduction owes to comments and suggestions by I. Polyakov. We strongly agree that introducing these water masses is useful for interpreting the results. For details on this point please refer to the revised manuscript. We also provide additional detail on halocline formation mechanisms in the first paragraph because I. Polyakov suggested that the ST method may capture the beginning of new halocline formation. We now think that interpreting the results in the light of the Rudels et al. (1996) mechanism supports this idea. For details on this point please refer to the revised manuscript and also to our response to the reviewer comments by I. Polyakov.

In the second paragraph of the revised introduction, we underline the importance of the halocline for the present and future of the Arctic Ocean. In the third paragraph, we combined the paragraphs describing the existing criteria to define halocline and CHL base. In the forth paragraph, we describe our original motivation for suggesting the stability (ST) method. Our original motivation was to use a threshold variable to define the halocline base which is more closely related to the role of the halocline as a stable layer which prevents warm Atlantic Water from reaching the SML and reducing sea ice than either the density ratio or the temperature difference. The analysis of individual profiles suggested by both reviewers indicates that the increased robustness of ST method derives mainly from the different search direction in the ST method

compared to the two existing methods: in the ST method we search upward instead of downward. This avoids isolated depth minima caused by near-surface features. We discussed this point in the revised version of our manuscript.

- It is my understanding that here, the authors equate the CHL base to the point of maximum stability. Wouldn't the point of maximum stability rather be within the "cline" you are considering, and not necessarily at its base? If I misunderstood, I would suggest the authors clarify their method and the physical reasoning behind it, as stated above.

We revised our methods section for clarity. As stated in lines 116f of the original submission, the stability threshold in the ST method was derived based on the density ratio threshold from the DR method. The original motivation behind devising the stability method was that a stable halocline prevents warm Atlantic Water from reaching the surface mixed layer, where it can either melt existing sea ice or prevent sea ice formation. We argue that stability is more closely related to vertical mixing than either the density ratio or the temperature difference.

- The manuscript still lacks an actual evaluation of the performance of each of the detection methods presented here. I like the large scales comparisons, but it lacks some quantitative estimates (which ideally would take into account the varying seasons and basins) and some idea of which method performs best. Because of the diversity of situations, different tests may lead to different rankings of the presented methods. And that is fine, as long as these various results are explicitly presented and discussed.

In order to evaluate the methods and to better understand the reason behind the different robustness, we investigated individual profiles as suggested by both reviewers. The analysis (please refer to the revised version for details) suggests that (a) the ST method captured the beginning new halocline formation directly underneath relatively fresh surface water in the Eurasian Basin, (b) the TD method overestimated the halocline depth in the Eurasian Basin, (c) in the Canadian Basin isolated

halocline base minima in the DR and the TD method occurred because of a layer of warm Pacific Water, and (d) the ST method slightly overestimated the depths of the halocline base in the Canada Basin. These findings are broadly consistent with summary statistics of halocline depth in Fig. 7 of the revised manuscript. In the Eurasian Basin, the ST method more frequently identifies a halocline base not far below the SML base compared to the DR method. Based on suggestions by I. Polyakov, we now interpret this as an indication of the ST method capturing the beginning of new halocline formation. The TD method overestimates the cold halocline base depth in the Eurasian Basin (Fig. 7b). In the Canada Basin (Fig. 7d), the DR method detected a halocline base shallower than 160 m for 10.2% of the profiles and the TD method for 3.5% of the profiles, indicative of isolated maxima due to the influence of near-surface warm Pacific water. These numbers differ from the numbers stated previously during the discussion in our response to your reviewer comment. This is because the numbers stated previously were accidentally based on a dataset which was meant for kriging, and from which outliers had been removed as described in the original version of the manuscript, and also because we omitted low resolution profiles form the UDASH data set as described in the revised manuscript. Using the correct data and omitting low resolution data from UDASH also changed Fig. 7 as stated in the track changes version of the manuscript in a note in the figure caption. Slightly increasing the stability threshold in the ST method may lead to a better match between the halocline base depths from the ST and the TD method by moving the halocline base estimated by the ST method upward and by decreasing the sensitivity of the ST method to new halocline formation.

- The organization of the manuscript lacks fluid connections, both in the introduction and in the presentation of the results. Subsections seem to be organized thematically but without a clear logical order. I would suggest the authors consider reorganizing the overall manuscript (introduction and results) and the abstract as follows:

1) Clear introduction of what SML, CHL, and cold halostad are, and why

they matter (as already done in part for the CHL).

2) Presenting the previous methods that have been used to define these layers, and underlining eventual (knowledge) gaps in these methods.

3) Introducing the new method, the physical reasoning behind its development, and the goal it aims to achieve.

4) Demonstrating how the results of the new method compare to results from previous methods (qualitatively and quantitatively, by adding some basin-wise statistics for example), and what are the gains of the new method.

We have revised the introduction, computed basin-wise statistics (see above), and now use individual profiles in our comparison of the results. For details, please refer to the revised version and/or the track changes version of our manuscript.

Minor comments:

In the introduction in general:

It would be good to improve logical connections between paragraphs. For example, L26-27: what is the connection between Atlantification discussions and CHL characteristics? You could finish the previous paragraph by commenting on possible changes in the strength of stratification within the CHL that could either boost or hinder further Atlantification.

We agree that it would be good to improve the logical connections between paragraphs. However, the first paragraph of the revised introduction is entirely devoted to introducing the different layers and water masses. Atlantification is now only mentioned in the second paragraph, which underlines the importance of these concepts. This provides a very nice and clear separation, but makes a smooth transition difficult.

In general, I would suggest reshaping your introduction as follows (as stated above):

- Defining the broad concepts of cold halocline layer and cold halostad

We followed this suggestion, but also introduce water masses such as PSW and PWW and discuss halocline formation mechanisms, especially the Rudels et al. (1996) mechanism, as explained in our response to your first major comment.

- Underlining their importance for our understanding of the present and future Arctic ocean characteristics

We now introduce layers and water masses in the first paragraph and comment on their importance in the second paragraph.

- Listing criteria for their definition used so far, and eventually pros and cons

We list the criteria. We found a comparison of halocline base depths computed with different methods in Bertosio et al. (2022). But as far as we know, there has been no systematic comparison of the pros and cons so far. We think that the devised methods are all very useful for the purpose they were designed for.

- Explaining clearly the motivation behind creating a new set of criteria, and the problem you aim to address

Regarding the halocline base, the existing thresholds are not as closely related to the role of the halocline as a stable layer which prevents mixing (although Section 2.2.4 of the original manuscript suggests that density ratio and stability are indeed related). But we consider this a feature of the methods, and not necessarily a problem. Each of the methods targets different aspects, which may be more or less relevant to the question at hand.

- Finally, introduce the organization of the manuscript (as you already do)

We revised the paragraph, and we also revised the rest of the manuscript.

L22–26: Remind the readers in 1-2 sentences what Atlantification is. In fact, you partly do so in the following sentence, but this should come earlier. And isn't ice loss a symptom/characteristic of Atlantification, too?

The revised sentence reads as follows: *Retreating sea ice, increased surface heat flux and the retreat of the halocline have been called atlantification of the Eurasian Basin (Polyakov et al., 2017).*

L38–39: Here, and generally throughout the manuscript, please define concepts as early as they appear. This sentence should come as you mention the density ratio, 2 lines above. In short, put the sentence line 37–38 at the end of this paragraph.

Thank you very much. We moved the sentence in lines 37-38 down, so that density ratio is introduced before mentioning studies which adopted the DR method. In the overview of halocline layers and waters in the first paragraph of the introduction, we provide short definitions of the Pacific Halocline Water (PHW) and the Lower Halocline Water (LHW) in parenthesis before expanding on details further below in the same paragraph.

L34–52: It would be best to reshape this series of 3 paragraphs into one, listing all existing used criteria to define the base of the CHL and eventually their pro and cons.

We combined the three paragraphs. Please refer to our response above regarding a discussion of the pros and cons already in the introduction. Thanks to the reviewers' suggestion to focus on individual profiles, we consider ourselves in a much better position to provide a more substantial discussion of the pros and cons in the results and the conclusion section of the revised manuscript.

L57: You already introduced this concept above when citing Bertosio (2020) and (2022). Would be best to merge the descriptions of upper and lower CHL into one paragraph here, keeping the relevant citations.

We merged the paragraphs and kept the citations. The relevant concepts are introduced in the first paragraph of the revised introduction.

L75: "Level III data": This processing level naming convention is rather opaque for unfamiliar readers. State more clearly what this entails (visual inspection, vertical interpolation, salinity spikes or bias corrections... etc)

The data processing for the ITP data is described by Krishfield et al. `http://www.whoi.edu/fileserver.do?id=35803&pt=2&p=41486`. Producing Level III

data included removal of corrupted data, corrections for the sensor response behavior, calibrations, and final screening of spurious outliers. We mention this in the revised version of our methods section.

L91: "Gaussian filter": as above, please introduce each processing method explicitly.

We revised the sentence as follows: *In order to reduce noise, the data was smoothed using a standard one-dimensional Gaussian filter (convolution with a Gaussian function, e.g. Deng and Cahill, 1993) with a standard deviation of 2 dbar and a truncation at ±10 dbar.*

Where the standard deviation and the truncation are in dbar instead of m because we have re-processed the data at the original vertical resolution without prior re-gridding and reduced the filter width and truncation.

Please note that Gaussian filters are a standard tool in signal processing (`https://en.wikipedia.org/wiki/Gaussian_filter`). Gaussian filters are generally more efficient at filtering high frequency noise compared to box filters (also known as running mean). Bourgain and Gascard (2001) used box filters of different widths for different variables.

L89: Is this reasonable for all profiles? Could some profiles, especially the oldest ones, have a vertical sampling of 5 to 10 m? If that is the case, please state so and briefly discuss why you think such a high-resolution interpolation is appropriate and reasonable. I would also suggest the authors consider a vertical resolution that is less fine and closer to the native profiles' vertical resolutions.

We reprocessed the data at the native resolution and we also omitted low-resolution profiles. In the revised version, only profiles with a vertical resolution finer than 2.5 dbar in the upper 300 m and a vertical resolution finer than 5 dbar elsewhere were used.

L139: it seems this sentence has grammar issues.

Yes, thank you. In the revised version, this sentence reads:

*A CHS will only be recognized by the algorithm if the vertical distance between the deeper stability maximum the first upper occurrence of that same stability*

*value is at least 50 m, and the difference of L between the lower stability maximum and the local minimum in the CHS is at least 0.2.*

L141:. . ."only the lowest of these layers is identified as a cold halostad." Why is that, physically?

The sentence in the original manuscript stated that in *rare cases, in which more than one layer fulfills these conditions, only the lowest of these layers is identified as a cold halostad*, although we had not encountered any such cases, and only now added a few lines of code to identify and count such cases. It turned out that no such case occurred. We revised the statement accordingly.

L169: 25th/75th percentile: This is an extremely stringent test. Can you explain why you took such a high threshold? As the other reviewer stated, it would be good to know how sensitive your results are to the "outlier" threshold you chose.

Outliers were defined as values outside the interval $[Q1 - 1.5\,IQR, Q3 + 1.5\,IQR]$, where $Q1$ is the first quartile, $Q3$ the third quartile, and $IQR = Q3 - Q1$ is the inter-quartile range. The section on kriging was omitted.

L254: In general, I would try to limit references to previous specific figures from previous papers. It is easier if you directly remind the readers what were the findings shown that figure through text, citing the source paper, in discussing your results in light of it.

This discussion was removed. The revised version contains updated figures in which we removed the panels showing maps of the SML .

L314: If you mean "comparatively low-salinity" then use "low salinity".

Yes, thank you very much. We followed your suggestion and corrected this expression in the revised introduction.

Fig. 2 and 7: I like these figures, and it would be great to have one of such plots for -when possible- each subregion and season. Even on one ITP, you could for example indicate when the buoys are in Canada vs. Makarov vs. Amundsen / Nansen basins.

Thank you very much for liking these figures. They were inspired by Polyakov et al. (2017). The revised manuscript contains revised versions of Figs. 2 and 7. Fig. 7 shows statistics for selected basins. We cited your reviewer comment suggesting to include basin-wide statistics at the end of the revised introduction.

Fig. 3: This is an interesting visualization, but it is rather under-used in the manuscript. I would suggest the author consider replacing it with plots presenting the vertical profiles, either in the introduction to present your various concepts (SML, upper and lower CHL, cold halostad), or in your results by grouping profiles in similar regions or seasons.

We analyzed individual profiles (please refer to the revised manuscript). Fig. 3 was replaced. Fig. 7 of the revised manuscript shows summary statistics. The figure is discussed in Sect. 3.2 of the revised manuscript.

Fig. 4 and 5: I still do not quite get the goal of using 2 interpolation methods (NN and kriging). If there is no other objective that showing the pan-Arctic results in 2 different ways, I would suggest the authors pick one of these methods and eliminate the other, in order to make the manuscript more fluid.

The kriging was eliminated.

Thank you very much again. We very much appreciate your comments. Please refer to the track changes version for a detailed description of all the revisions. Because of the substantial revisions in almost all sections, this response is still limited to summarizing and outlining major changes. We have re-written parts of the methods section for clarity, revised the results section based on the very constructive comments and suggestions by both reviewers, and also adapted the title, the abstract and the summary section accordingly. We did not attempt to describe and motivate each of the individual changes in this response. Instead, we marked the changes in the track changes version. In some cases, including cases in which we noted and removed errors, we also motivated and explained changes in the track changes version. In a few places, we changed the draft version of the revised introduction that was included in the original response to your reviewer comment. However, we completely kept track only of the changes compared to the original manuscript and not to this draft. We very much appreciate your effort in reviewing this technical note

and consider the suggestions from the reviews a key contribution to the revised manuscript.

**Technical note: Determining Arctic Ocean [I.P] halocline and cold halostad [I.P] depths based on vertical stability**

Enrico P. Metzner[1] and Marc Salzmann[1]

[1]Institute for Meteorology, Universität Leipzig, Leipzig, Germany,

**Correspondence:** Enrico P. Metzner (enrico.metzner@uni-leipzig.de)

**Abstract.** The Arctic Ocean [I.P] halocline [I.P] separates the cold surface mixed layer (SML) from the underlying warm Atlantic Water (AW), and thus provides a precondition for sea ice formation. Here, we introduce a new method in which the [I.P]halocline base depth is diagnosed from vertical stability and compare it to two existing methods. [M.A.]Our

5 main motivation for diagnosing the halocline base depth based on vertical stability was that vertical stability [M.A.]is closely related to vertical mixing and heat exchange[M.A.], and thus also to the role of the halocline in preventing vertical heat exchange and thereby protecting sea ice from warm AW. The second goal was to provide a particularly robust method. When applied to measurements from ice-tethered profilers, ships, and moorings, the new method for estimating the [I.P]halocline base depth provides robust results with few artifacts. Comparatively large differences between [I.P]the methods for detecting the [I.P]halocline base depth were found in [I.P]warm AW inflow regions [I.P]
[revised manuscript text omitted]
 = \left( \alpha \nabla_z \theta_{\text{pot}} \right) / (\beta \nabla_z S)$ **Please note: The track changes package we use does not handle subscripts correctly. The subscript "pot" was removed and not added.** with potential temperature $\theta_{\text{pot}}$ in $°C$**Please note: This does not make difference for the gradient**, salinity
145 $S$ in the practical salinity scale and depth $z$ in m. $\alpha = -\rho^{-1}(\partial\rho/\partial\theta_{\text{pot}})$**Please note: Sign of $\alpha$ corrected** and $\beta = \rho^{-1}(\partial\rho/\partial S)$ are the thermal expansion coefficient and the haline contraction coefficient, respectively. Bourgain and Gascard (2011) empirically estimated that searching downward for the depth, in which $R_\rho$ exceeds 0.05, provides a reasonable estimate for the base of the [I.P]halocline. The search starts at the base of the SML (here determined as described in Sect. 2.4 below), which is defined to be the top of the halocline layer
150 . [M.A.]If the density ratio threshold is exceeded already directly at the base of the SML, then no halocline was detected for the corresponding profile. Such profiles are excluded when computing statistics of halocline base depths. Similarly to Bourgain and Gascard (2011), we smoothed the $S$ and $\theta$ prior to computing the density ratio as explained in Sect. 2.5.

155 #### 2.1.2 [I.P]Temperature difference (TD) method

The temperature difference (TD) method (Metzner et al., 2020) uses the difference $\Delta T$ between the ocean temperature $T$ and the sea water freezing temperature $T_{\text{freeze}}$ [M.A.]to estimate the cold halocline base depth. The freezing temperature was calculated from Gill (1982). [M.A and I.P]Searching downward, starting at the SML base, the base of the [I.P]halocline was calculated as the depth, in which $\Delta T$ first exceeds 1K. This threshold was estimated to be high enough, that the "cold"
160 core of the [I.P]cold halocline layer is detectable, and low enough to separate the CHL from the AW with

core temperature approximately $1.5\,°C$ to $3\,°C$ ($T_{\text{freeze}} \approx -2\,°C$ leads to $\Delta T \approx 3.5\,°C$ to $5\,°C$ at the core). In cases in which the temperature threshold was first exceeded in a depth [I.P]shallower than $80\,$m, the search was continued $0.5\,$m below this depth. [I.P]If the temperature threshold was exceeded already at the SML base, no halocline was detected. The algorithm was applied to smoothed temperature data (see Sect. 2.5).

**2.1.3 [I.P]Stability (ST) method**

The new stability (ST) method prescribes a threshold for the local vertical stability in order to estimate the halocline base depth. Vertical stability is more closely related to vertical mixing than either the density ratio or the temperature difference. Because the search direction was found to affect the robustness of the method, and because stability is decreasing with depth between the AW and the core of the halocline, we search upward instead of downward for the stability threshold. The stability was computed from $L = \log_{10}(N^2)$, where $N = \sqrt{-(g/\rho)\,(\partial\rho/\partial z)}$ is the Brunt-Väisälä-frequency (with density computed from pressure, smoothed $S$, and $\theta$ as described below). The stability threshold was approximated based on the density ratio threshold $R_\rho = 0.05$ assuming an approximately constant salinity gradient near the halocline base in the LHW. It was derived starting from the following relationship:

$$\rho^{-1}\nabla_z\rho = \beta\nabla_z S - \alpha\nabla_z\theta_{\text{pot}} = \beta\nabla_z S (1 - R_\rho) \tag{1}$$

With stable $\beta = (7.82 \pm 0.03) \cdot 10^{-4}$ over a wide range of temperature, salinity and pressure values ($-1.2\dots2.0\,°C$, $32\dots37$, $50\dots350\,$dbar) and the salinity gradient in $\text{m}^{-1}$:

$$L = \log_{10}(-\nabla_z S) - 2.137 \pm 0.002 \tag{2}$$

Expecting the salinity gradient to be around $0.01\,\text{m}^{-1}$ near the base of the halocline, the resulting stability threshold should be $L \approx -4.14$.

This threshold is searched from $600\,$m or at the lowest point (at least $500\,$m deep outside the shallower regions according to the conditions for including profiles in the analyses described below in Sect. 2.5) to the surface, as no CHL base was observed deeper than that. Seldom, the first estimate is in warm [I.P]AW at $T > 0°C$. In such cases, a second search for the stability threshold is started slightly above. [M.A.]If the stability threshold is never exceeded or only exceeded where $T > 0°C$ in a given profile, then no halocline base was detected for this profile. ~~When the maximum stability is lower than the threshold, which can happen in shallow regions, the temperature difference $\Delta T$ at the base of the SML was checked against the threshold from the TD method. If $\Delta T$ is greater than 2 K, the water mass was considered Atlantic water and the CHL thickness was set to zero. Otherwise, the algorithm failed and a fill value was returned. This method will be further referred to as stability (ST) method.~~ **Please note: We simplified the explanation. Whether the halocline thickness was set to zero or treated as a missing value does not make a difference because in the end, both are treated as "no halocline detected".**

**2.2 Cold hHalostad (CHS) boundary and center estimates**

[I.P.]A CHS is formed by PWW in the Canada Basin and also by melt water off the eastern coast of Greenland and Svalbard.

195 Compared to the CHL in the Eurasian Basin, a CHS is characterized by a smaller salinity gradient because of the different water origins. As demonstrated below, this leads to one local stability maximum at the top of the CHS (at the transition between SML or PSW and PWW) and a second stability maximum associated with the transition between CHS and the LHW. The stability minimum between these two local stability maxima is associated to the CHS. Therefore, as a first condition for identifying a CHS, we require that more than one local stability maximum must be present between the base of the SML and

200 the base of the halcoline as identified by the ST algorithm described above. **Please note: We failed to mention the following point in our original manuscript. The operation described below is applied in addition to the requirement that the vertical extent of the local minimum must be at least 50 m** [M.A.]Because the stability profiles computed from temperature and salinity observations contain small scale fluctuations even after smoothing the $S$ and $\theta$ data that is used for computing density as described below, we identify local maxima by first computing a "moving" stability maximum for a 50 m vertical box

205 surrounding each observation. This moving stability maximum is computed from $L_m(z) = \max\left(L(z') \text{ for } |z' - z| < 25 \,\mathrm{m}\right)$, where $z$ is depth. This moving maximum operation was defined in analogy to a moving average. The result is a profile of stability maxima $L_m(z)$ with few local maxima. [I.P.]We then compute the mean of the lower stability maximum, which is associated with the transition between CHS and LHW, and the stability minimum between the upper and the lower stability maximum, which is associated with the CHS.In case of multiple stability maxima in the vertical profile, the thickness and

210 the depth of the cold halostad are estimated. The cold halostad separates the CHL into an upper and a lower branch. It is associated with a stability minimum between two stability maxima. In order to estimate the thickness of the cold halostad, we first compute the mean of the stability minimum and the stability maximum associated with the lower branch of the CHL. This value is used [I.P.]as a threshold to define the [I.P.]upper and the lower boundary of the CHSthe boundaries of the cold halostad. The depth of the [I.P.]center of the CHScold halostad is [I.P.]definedestimated as the mean of the upper and lower bound[I.P.]ary of the

215 CHS. A CHScold halostad will only be recognized by the algorithm if the vertical distance between [M.A.]the deeper stability maximum the first upper occurrence of that same stability value is at least 50 m, and [M.A.]the difference of $L$ between the lower stability maximum and the local minimum in the CHS is at least 0.2this stability layer has at least a relative depth of 0.2 in $\log_{10}(N^2)$. [M.A.]With this definition, we never identified more than a single CHS per profile.In rare cases, in which more than one layer fulfills these conditions, only the lowest of these layers is identified as a cold halostad.

220 ## 2.3 [I.P.]Occurrence frequency of the CHL and cold halostad

**Please note: We now explain more explicitly what it means that no halocline was detected for a given profile above, omitting technical details, especially the treatment of rounding errors in floating point operations. Based on this, the occurrence frequency is now defined below in the results section.** [I.P.]Occurrence frequencies of the CHL were computed for each method individually based on whether the criteria for estimating the CHL base depth were satisfied. Furthermore, a

225 lower threshold of 0.001 m was used for the CHL thickness, below which no distinctive layer is assumed to exist. Cases in

which either the criteria was not satisfied or else the thickness was computed to be less than $0.001\,\mathrm{m}$ were the classified as "no CHL". This includes cases in which the CHL and the SML bottom coincide within numerical accuracy and cases in which we initially computed a value for the CHL base depth that was smaller than the value for SML base depth.

[I.P]The depth of the cold halostad layer was computed whenever data was available and the conditions for identifying a cold halostad layer as described above were fulfilled. The occurrence frequency for the cold halostad layer was only computed for a region in the Canada Basin. Outside this Canada Basin region, the occurrence frequency was not computed due to the sparsity of observations in these regions.

**2.4 SML depth estimate**

The SML depth was estimated by a change in potential density of $0.125\,\mathrm{kg\,m^{-3}}$ at the surface as in Polyakov et al. (2017). In cases, in which a CHL is detected, the depth of the SML corresponds to the top of the CHL. Potential density was computed from smoothed $S$ and $\theta$.

[M.A and I.P]**2.3 Kriging**

[M.A and I.P]The estimates of the SML thickness, the CHL base depth and the occurrence frequency of the CHL are inter- and extrapolated with kriging. Kriging (e.g. Deutsch, 1994) is a tool from geostatistics for interpolation, extrapolation and smoothing at locations without direct measurements by data from surrounding observations. Those observations are weighted together depending on the local variability estimated by a multidimensional autocovariance function. The latter is approximated with a so called semi-variogram model. In this analysis only the spherical semi-variogram model is used (Mert and Dag, 2017; Reshid, 2019). The weights take into account the clustering of the observations as it uses the inverse of the correlation-distance matrix between input points and the correlation-distance vector between input points and the point of interest. All inter-/extrapolation weights are calculated from the location and distribution of given information without prior knowledge of ocean currents.

[M.A and I.P]Here, kriging was performed on an azimuthal-equidistant projection. The North Pole was used as the central reference point. $0°\,\mathrm{E}$, $90°\,\mathrm{E}$, $180°\,\mathrm{E}$ and $90°\,\mathrm{W}$ are on negative y-axis, positive x-axis, positive y-axis and negative x-axis respectively. The resolution is $0.46875°$ or roughly $52\,\mathrm{km}$ at the North Pole. The year was equally separated into 12 periods to simplify calculations and time averaging afterwards. This lead to $36\mathrm{x}12 = 432$ time steps and a temporal resolution of about $30.4\,\mathrm{days}$.

[M.A and I.P]Each quantity was prepared for kriging in three steps:

1. [M.A and I.P]A spatial trend of the time mean was computed.

2. [M.A and I.P]Outliers were removed for each season separately. Data values were identified as outliers if the difference to the spatial trend was lower (higher) than the $25^{\mathrm{th}}(75^{\mathrm{th}})$ percentile decreased (increased) by 1.5 times the inter quartile range.

3. [M.A and I.P]The spatial trend was subtracted from the raw data. The resulting anomalies were normalized to ensure a nearly constant variance for the entire Arctic.

[Figure]

**Figure 1.** Please note: Revised figure. The figure differs from the corresponding figure in our response to the reviewer comment by I. P. because we omitted low resolution profiles form the UDASH data set as described in the revised manuscript. [I.P]L ocations of observations for each season [I.P](starting with MAM for March, April, and May) with blue dots for UDASH profiles and red dots for ITP profiles **(a–d)**. Temporal coverage for UDASH **(e)** and ITP **(f)** observations.

[M.A and I.P]

**2.5 Data and preprocessing**

[T]emperature and salinity observations were taken from the ice tethered profiler (ITP) project (Krishfield et al., 2008; Toole et al., 2011) and the Unified Database for Arctic and Subarctic Hydrography (UDASH, Behrendt et al., 2018). The ITPs measured temperature, salinity, and pressure twice a day while drifting with the ice floe they were tethered to. [Moved down, Rev. #2][M.A.]Data processing for the ITP data is described by Krishfield et al. (http://www.whoi.edu/fileserver.do?id=35803&pt=2&p=41486). Here, we used processed ITP Level III data. Producing Level III data included removal of corrupted data, corrections for the sensor response behavior, calibrations, and final screening of spurious outliers. [Moved here, Rev. #2]ITPs deployed in the Arctic Ocean before 2018 were included here. [I.P]The vertical resolution for ITP level III data is 1±0.1 dbar. The accuracy of the sensors used for the ITP observations is 0.002°C for temperature and 0.002 for salinity (Polyakov et al., 2017). [I.P] [Moved here, Rev. #2]The

UDASH data set contains data from ships, ice-tethered profilers, profiling floats and other platforms (Behrendt et al., 2018). rom the UDASH data set, nly profiles, for which both temperature and salinity were available, were analyzed here. [I.P]Furthermore, only profiles with a vertical resolution finer than 2.5 dbar in the upper 300 m and a vertical resolution finer than 5 dbar elsewhere were used[Moved up, Rev. #2] We also required that the deepest point in a profile must reach at least 500 m or 90% of the basin depth. This choice addresses the issue of potential sampling biases due to limited vertical extend of the observed profiles. [I.P]Regions shallower than 100 m were always excluded from the analysis. Bathymetry data was taken from the General Bathymetric Chart of the Oceans (GEBCO) dataset (GEBCO Bathymetric Compilation Group 2021, 2021). [I.P]This filtering left a total of 43715 ITP and 62012 UDASH profiles. Fig. 1 provides an overview of the spatio-temporal coverage of the data. **Please note: This fairly small difference between the total number of profiles compared to the original manuscript stems from a compensation between additional filtering (decreasing the number of profiles analyzed) and no longer binning the ITP data to two-day intervals as before (increasing the number of profiles analyzed).** Most measurements are concentrated in the Barents Sea and only few were taken in the Central Arctic during winter. For the East Siberian Sea and the interior of the Laptev Sea, no data was available for winter and spring. Salinity was given in the practical salinity scale.

Depth was computed from pressure using the hydrostatic equation. Density was computed based on salinity, temperature, and pressure.**Please note: All observations used here were reported on pressure levels. Mentioning iterations was an error. An iterative approach was only applied to model data, which we did not use here.** [I.P][M.A.]In order to reduce noise, $S$, $T$, and/or $\theta$ were smoothed using a standard one-dimensional Gaussian filter (convolution with a Gaussian function, e.g. Deng and Cahill, 1993) with a standard deviation of 2 dbar and a truncation at $\pm10$ dbar. When using thresholds to estimate the SML or CHL base depth, variables were linearly interpolated between two adjacent depths. Consequently, the SML or CHL base can be located between two vertical [I.P]observation points and the SML and CHL base depths do not necessarily have to coincide with the depths of the [I.P]observations.

[Figure]

**Figure 2.** **Please note: Revised figure using level III data without regridding.** ITP-74 lLocation of measurements **(a)** and time series of (a) temperature **(b)**, , (b) salinity **(c)**, and (c) vertical stability **(d)**along the path of ITP 74. The circle and the cross in (a) mark the beginning and the end of the ITP-74 track, respectively. The colored lines in (b–d) are the base of the SML (black) and the [I.P]halocline (HCL)CHL base depths derived by the DR method (blue), the TD method (red) and the ST method (white). Salinity is given in the practical salinity scale. [M.A and I.P]Individual profiles at the location of the wedge symbols (∧) below the x-axis in (b) are shown in Fig. 3. Profiles that started below 15 m were excluded (only) in this figure (but nowhere else), because this increased readability by reducing the effect of noise in determining the SML base without affecting the overall result.

**3   Results**

**3.1   [I.P]Comparison of methods for deriving haloclineCHL base depth using case studies**

In Figure 2, we compares three different methods for determining the [I.P]haloclineCHL base depth for ITP-74. Starting from
305   the Laptev Sea in SeptemberOctober 20132, ITP-74 drifted across the Central Arctic, almost reaching the East Greenland Sea [M.A and I.P](Fig. 2a). Until May 2014, [M.A and I.P]Fig. 2b–d showsthere is evidence of a well-defined and stably stratified CHL [M.A and I.P]below the SML. The vertical stratification observed by ITP-74 prior to May 2014 and the performance of the three methods for determining the halocline base depth are further analyzed in an individual profile from this period in Fig. 3. Individual profiles of salinity and temperature for 13 January 2014 in Fig. 3a show the base of the SML at about ∼30 m.
310   Between the SML base and ∼80 m a strong salinity gradient and temperatures close to the freezing point indicate a well defined CHL, which is ∼60 m thick. Between the CHL and the AW, temperature and salinity increase in the LHW. Below ∼170 m warm and saline AW is found (please note that at ∼170 m, the temperature and salinity slopes in Fig. 3a both change). Figs 3b–d show the density ratio, the temperature difference, and the stability for the observations on 13 January 2014 together

[Figure]

**Figure 3.** **Please note: Figure added based on suggestion by both reviewers. In our response to the reviewer comment by M. A. Figure 3h accidentally showed a dash-dotted instead of a dotted line. This was corrected.**[M.A and I.P]Temperature $T$ and salinity $S$, density ratio, temperature difference, and stability from ITP-74 before winter convection on 13 January 2014 **(a–d)**, during winter convection on 20 May 2014 **(e–h)** and after winter convection on 15 July 2014 **(i–l)**. Vertical dashed lines indicated threshold values. Horizontal dashed lines indicate the halocline base determined by the three different methods. Dotted lines indicate the SML base. Dash-dotted lines in (f) and (g) indicate that a threshold for identifying the halocline base was exceeded at the SML base.

[revised manuscript text omitted]

405 The underlying grid uses an azimuthal-equidistant projection with ∼0.47° resolution at the North Pole. [M.A.] In addition to these

[Figure]

**Figure 6.** Please note: Revised and combined figure, based on figures 5 and 6 of the original manuscript. Kriging and SML maps are now excluded. We also excluded regions shallower than 100 m. The figure differs from the corresponding figure in our response to the reviewer comment by I. P. because we omitted low-resolution profiles form the UDASH data set as described in the revised manuscript. [M.A and I.P]Halocline base depth (BD) derived by [M.A and I.P]the density-ratio [M.A and I.P](DR) algorithm [M.A and I.P]**(a)**, [M.A and I.P]the temperature-difference [M.A and I.P](TD) algorithm [M.A and I.P]**(b)**, and [M.A and I.P] the new stability [M.A and I.P](ST) algorithm [M.A and I.P]**(
[revised manuscript text omitted]

**3.3 [M.A and I.P]**

[Figure]

**Figure RM5. Please note: This figure was revised and combined with the preceding figure.**[M.A and I.P] ~~Comparison of (left) simple nearest-neighbor averaging to (center) kriging results with (right) difference plot for (a–c) thickness of SML and CHL base depth derived by (d–f) DR method, (g–i) TD method and (j–l) the new ST method. Points where the relative occurrence frequency of the CHL is below 1% were masked out in (d–c). Hatching indicates regions where the CHL occurrence frequency is below 25%. The SML depth in the East Siberian Sea and the interior of the Laptev Sea is based on observations in summer and fall.~~

[M.A and I.P]
475

[M.A and I.P]
480

M.A and I.P Although kriging acts to smooth the field, Fig. RM5b still shows some small scale variability in the Central Arctic, which may result from relatively sparse observations in winter and spring (Fig. ??) in combination with a fairly pronounced seasonal cycle (Fig. RM6).

M.A and I.P The time mean CHL base depths determined from the DR method (Fig. RM5d, e), the TD method (Fig. RM5g, h), and the ST method (Fig. RM5j, k) show a similar overall pattern. Comparatively deep time mean CHL bases below 200 m are found in the Canada Basin. The CHL base depth decreases as one moves from the Canada Basin through the central Amerasian Basin and then across the North Pole into the Eurasian Basin. This overall pattern is consistent with figures 4g and h of Polyakov et al. (2018). This CHL base depth gradient is roughly opposite to the gradient of the SML depth.

M.A and I.P There are, however, several notable differences between the methods to determine the CHL base depth. For example, the TD method yields somewhat larger CHL base depths over the Central Arctic compared to the DR and the ST method. A particularly striking difference in Fig. RM5 is found for the CHL base depth in the Barents Sea. While CHL occurrence frequencies in the Barents Sea are fairly low for all three methods, they differ notably, depending on which method is used to detect the CHL base (Fig. 6). For the DR method, the CHL base depth decreases from the Barents Sea toward the Central Arctic. This contrasts the finding by Steele et al. (1995) that the CHL forms in the Barents Sea and deepens where Atlantic water sinks at the slopes north of the Barents Sea into the Eurasian Basin. On the other hand, complex situations similar to the one observed by ITP 74 are difficult to capture by methods that to some extent rely on simplification. We nevertheless chose to include the Atlantic and the Barents Sea in our comparison, in part because it helps to highlight problems, which with increasing atlantification may also occur in other regions.

M.A and I.P Furthermore, as indicated above, the CHL occurrence frequency in the Barents Sea is generally low. Especially for the TD method, a CHL is rarely detected in the Barents Sea (Fig. 6b). Because the CHL base depth is only computed for cases in which a CHL is detected, the time mean SML depth in Fig. RM5 often exceeds the time mean depths of the CHL in the Atlantic and also the Barents Sea.

M.A and I.P The kriged CHL base maps (Figures RM5e, h, k) are smoother compared to the kriged SML map (Fig. RM5b), in part because of a smaller seasonal cycle (Fig. RM6). Artifacts in the DR method are partially filtered out here because spurious depth minima in the CHL base often reach the SML, so that these cases are not taken into account in computing the mean CHL base depths.

**3.4 M.A and I.P SML and CHL base depth seasonality**

M.A and I.P The seasonality of the SML and the CHL base depths are investigated in Fig. RM6. Insufficient data availability is an issue especially in the East Siberian Sea and the interior of the Laptev Sea, where both the seasonal cycle of the SML depth and the CHL base depth are most likely underestimated because of the absence of observations in winter and spring. The SML base depth varies more strongly with season than the CHL base depths for the East Greenland Sea, the Barents Sea, and the Atlantic. Together with the different spatial gradients in annual mean depths, this presumably makes the CHL in these regions more prone to ventilation events in winter, such as the ones investigated by Polyakov et al. (2018). This stronger seasonal cycle

[Figure]

**Figure 7.** **Please note: Figure added based on suggestion by M.A. This plots differ from the corresponding plots in our response to the reviewer comment by M. A. mainly because the version of the plots in the discussion was accidentally based on a dataset which was meant for kriging, and from which outliers had been removed as described in the original version of the manuscript, and also because we omitted low resolution profiles form the UDASH data set as described in the revised manuscript.**[M.A.] Map showing elliptical areas over the Canada Basin (purple), the Makarov Basin (green) and the Eurasian Basin (dark red) and ocean floor depth (grey shading) **(a)**. Relative frequency of halocline (HCL) base depth determined with the density ratio (DR), temperature difference (DR) and the stability (ST) method for the elliptical areas over the Eurasian Basin **(b)**, the Makarov Basin **(c)**, and the Canada Basin **(d)**.

[revised manuscript text omitted]

---

## Author Response (AR2)

**Response to reviewer comments by Igor Polyakov**

Thank you very much for your very insightful second review and very constructive comments. We repeat your comments (using a dark red font color) below. Our responses are given in black font.

The authors made substantial revision of the original manuscript following the reviewers' comments. However, there are still several issues that need the authors' attention. They are related to the application of the methods and the quality of data processing and analysis. I think this study still requires major revision.

Comments:

1. Lines 3–4: The sentence is hard to read. Needs editing.

   We split the sentence in two and replaced "based on" by "depending on" in order to avoid the awkward combination of "base" and "based on". We also included the statement "unlike parameters previously used to detect the halocline base" in order to emphasize the contrast to existing methods. The revised version reads:

   *Our main motivation for diagnosing the halocline base depth depending on vertical stability was that vertical stability is closely related to vertical mixing and heat exchange. Unlike parameters previously used to detect the halocline base, vertical stability is a crucial parameter for determining whether the halocline can prevent vertical heat exchange and protect sea ice from warm AW.*

   In an attempt to provide a more balanced discussion of pros and cons, we added the following sentence to the section describing the stability method (Sect. 2.3.1):

*However, while vertical stability is closely related to vertical heat exchange and to the role of the halocline for protecting sea ice, the density ratio is more directly related to the original definition of a halocline.*

2. Abstract: You can drop all abbreviations from the abstract if they are used only once. They will need to be defined in the text anyway.

Done. Thank you. We only retained AW for Atlantic Water because it is used in the abstract.

3. Lines 6-7: These two sentences can be merged.

Thank you for your suggestion. We omitted the first sentence.

4. Lines 10-13: The last two sentences do not belong to the end of Abstract. The authors may use GCM argument (ll. 9-10) to close this short discussion.

Thank you very much. We very much appreciate your suggestion. We moved both sentences up. We also slightly modified the new last sentence of the abstract because we noted an ambiguity. The original sentence was:

*Comparatively large differences between the methods for detecting the halocline base depth were found in warm AW inflow regions for which climate models predict increased net surface energy fluxes from the ocean to the atmosphere, suggesting that these regions may be particularly sensitive to a halocline retreat.*

In this original sentence, it was not entirely clear that the "*suggesting that*" refers to the analysis of model results mentioned in the second part of the sentence, and not to the large differences mentioned in the first part of the sentence. Therefore, we adapted the sentence as follows:

*Comparatively large differences between the methods for detecting the halocline base depth were found in warm AW inflow regions for which climate models predict a halocline thinning and increased net surface energy fluxes from the ocean to the atmosphere.*

5. Line 15. "Inflow regions" instead of "inflow"?

   Yes, thank you. We replaced "inflow" by "inflow regions".

6. Discussion of halocline formation (e.g. ll. 26-30 and throughout the text). Now the text reads as the freshwater capping is involved in and helps the halocline formation. It is partially true since the freshwater is the source for the density gradient in the halocline. But the formation of the halocline is driven by a) horizontal advection (injection) of denser water from shelves in winter and b) vertical mixing (overcoming stability of the surface fresher layer) due to winter sea ice formation and brine rejection. Thus, the physical drivers overcome stability of the water column to form the halocline.

   Thank you very much. Based on a comments by both reviewers, we modified our discussion regarding the the interplay between convective homogenization and capping in the introduction of the manuscript. The revised part reads:

   *Based on data from the Oden 1991 cruise, Rudels et al. (1996) found that new halocline formation was initiated by the advection of relatively fresh shelf waters near the surface above denser and more saline water below, when the advection of the fresh water limited winter convection. Alkire et al. (2017) and Rudels et al. (2004) argued that convective homogenization and capping by fresh water due to sea ice melting in the inflow from the Fram Strait and the Barents Sea can transform AW into halocline water. Rudels et al. (2004) stressed that melting provides a precondition for sea ice formation and convective homogenization during the following winter.*

   We decided to omit the following sentence, which we deem less relevant in the light of our extended discussion on the roles of melt water:

   *Steele and Boyd (1998) argued that seasonal capping by melt water in summer may not be overly important for insulating the SML from relatively warm AW.*

   Throughout the manuscript, we now refer to the mechanisms that involve

convective homogenization and capping as "convective homogenization and capping" instead of simply "capping".

7. Lines 36-37: Logic is broken in the LHW part, the sentence needs editing.

We added "not only in the Eurasian Basin but" to the LHW part of the sentence and also replaced the "but" in the beginning of this part by "while". However, we are not sure if this addresses your concern. If not, could you please elaborate? Figure 5 of Polyakov et al. (2018) shows LHW below PHW extending to the Bering Shelf. Based on Anderson et al. (2013), we state that this LHW water is of Atlantic origin. Did we misinterpret this? The revised sentence reads:

*The PHW in the Canada Basin originates from Pacific Water inflow, which is modified on the Chukchi Sea Shelf, while the LHW is of Atlantic origin not only in the Eurasian Basin but also in the Canada Basin (e.g. Anderson et al., 2013).*

8. Lines 44-46: Editing is needed. It would be helpful if the authors would illustrate statements like that with figures.

We changed the lines as follows:

*Below, we argue that an increase in salinity associated with the PWW and another increase associated with the LHW results in two distinct local stability maxima between the surface and the LHW (compare also schematic in Supplement 1 and profiles in Supplement 2).*

referring to Figures in two newly added supplements. Below the next sentence (referring to the absence of the second stability maximum in the CHL in the Eurasian Basin), we added the sentence:

*This allows us to identify the CHS.*

The sentence referring to the CHS formed by Greenland melt water was moved to the end of the section.

9. Line 50: "ist".

Thank you. We deleted the "t", translating the German word "ist" (meaning "is") into English.

10. Line 59, The authors may want to add reference to Shimadat etal 2006: K. Shimada, T. Kamoshida, M. Itoh, S. Nishino, E. Carmack, F. McLaughlin, S. Zimmermann, A. Proshutinsky, Pacific Ocean inflow: Influence on catastrophic reduction of sea ice cover in the Arctic Ocean. Geophys. Res. Lett. 33, L08605 (2006).

Done.

11. Line 98: This referencing to reviewers is not standard. I suggest to cut them.

We shortened the paragraph containing the manuscript outline as suggested by M. Athanase and and also cut the references to the reviews. But we did like the idea of citing the reviews in the text, and had initially found numerous arguments to defend our choice of citing the reviews.

12. Line 104: I suggest to change "In a halocline" to "in the Arctic halocline" and "must be" to "is".

Thank you. Done.

13. Section 2.2. I suggest the authors illustrate these features using vertical profiles of temperature, salinity, and density. Means would probably work best.

In Section 2.2, we now refer to the schematic showing salinity and stability profiles (Supplement 1) and to profiles based on observations (Supplement 2) which we added in response to your comment #8 above. The salinity profile in the schematic is qualitatively similar to the profiles shown in figure 2 of Shimada et al. (2005). The salinity and stability profiles in the schematic are also qualitatively similar to the corresponding profiles in Figure 5a and d except for the layer of Pacific Summer

Water. They are also qualitatively similar to the corresponding profiles in Figure 5e and h except for the SML.

14. Line 152: "Halocline", not "halcoline" (but sounds beautiful).

Thank you very much. We replaced "halcoline" by "halocline". Although there appears to be no obvious meaning of the word halcoline, we hope that it may eventually be adapted for some purpose. In this event, any credit would have to be bestowed on Aagaard et al. (1981) who have discovered the word already more than 40 years before us (`https://doi.org/10.1016/0198-0149(81)90115-1`).

15. Line 155: Please show this method using a profile.

We added Supplement 3 to explain the method using an example profile. Figure S3b illustrates the usage of the "moving maximum" operator.

16. Lines 156-163: Please show examples (profiles).

Supplement 3 contains three panels illustrating steps to determine the CHS bounds and center depth.

17. Line 167: Define "smoothed".

We added the following sentence in line 163 of the revised manuscript (line 187 of the track changes version):

*The smoothing was performed using a Gaussian filter as described in the next section.*

18. Line 176: The authors need to cite the source, not our paper.

The revised statement reads as follows:

*The accuracy of the sensors used for the ITP observations has been cited as $0.002°C$ for temperature and $0.002$ for salinity (Polyakov et al., 2017)*

*in agreement with an assessment by the manufacturer (Janzen et al., 2016). For temperature, this accuracy range is supported by Wong et al. (2023a). For salinity, larger biases can arise due to sensor shift on longer time scales, depending on the manufacturing date of the sensor (Wong et al., 2023b)*

Any further comments you may have on this are welcome.

19. Line 2003: Please define "slopes ... change". They change everywhere, so the sentence is not clear.

    We changed

    *please note that at $\sim170\,$m, the temperature and salinity slopes in Fig. 3a both change*

    to

    *please note the kink in the temperature and salinity profiles at $\sim170\,$m in Fig. 3a*

20. Lines 203-206: This is a way too long sentence which can be easily split for 2-3.

    The sentence starting in line 203 was split into two sentences.

21. Line 212: July 15 is not "After July".

    Thank you. We replaced "After" by "In".

22. Figure 3: This new Figure and its analysis are my major concerns.

    a. Since density is used for the analysis, I suggest it is also shown.

       Figures 3 and 5 of the revised manuscript include profiles of the potential density anomaly.

b. I see a problem with the salinity profiles which show local minima just above the high gradient areas. It is a well-known problem and comes from data processing. It creates artificial instabilities (higher over lower salinity layers). I think that is partially the reason for the high level of noise in the density ratio profiles. This problem may affect all authors' results so it should be addressed properly. There are several ways to reduce the problem. One way is to use raw temperature and conductivity and use local correlations to better adjust the profiles. Probably, it will be prohibitively expensive time-/effort-wise. Another one is to apply smoothing, but it requires a lot of considerations. I suggest the authors consider that, fix the problem in the salinity profiles and describe it carefully in the text.

Figures in Supplement 4 and Supplement 6 correspond to Figures 3 and 5. But they are based on Level I instead of Level III data. We also found spurious halocline base depth minima in the DR and the TD method when analyzing data from coupled climate models (Metzner et al., 2020). Because there is no data assimilation involved in these model runs, these model results are not influenced by such data processing problems. We also argue that the spurious depth minima in Figures 4 and 5 are related to a layer of warm PWW. Such a layer may explain spurious detections of halocline base depth minima even in the absence of noise.

Please note that in Supplement 5 we included maps showing the locations of the profiles in Figure 3 together with maps of sea ice concentration based on Spreen et al. (2008).

c. I am also wondering why the density ratio profiles change the sign from negative to positive within 60-170m whereas both temperature and salinity (as well as alpha and beta) are monotonic. I think within this depth range the density ratio should be positive. Please check. After fixing the problems described in b-c, the profiles and definitions (and further results!) may look differently.

The x-axis in the revised Figure 3c and h use cubic scaling (as explained in our answer to your next point). This makes it much easier to see where the density ratio changes sign. Where the gradients in Figure

3a are clearly positive and monotonic, the density ratio does not change sign. Isolated instances of negative density ratios do not cause the density ratio threshold to be exceeded because the density ratio threshold is an upper threshold.

d. The authors may consider using log axis for the density ratio to show the 0.05 value.

We now use a cubic scaling $(x^3)$ for the x-axis instead of log(x). Log(x) changes sign and is undefined for negative values. Because log(x) changes sign, one cannot simply compute the absolute. Using log(x), one ends up with positive and negative values on both sides of the $x = 0$. A cubic scaling preserves the sign.

e. (f-h) Why SML depth is at 130m? Density profile would help.

Figures 3c, h, m in the revised manuscript show the potential density anomaly. Vertical dotted lines show the surface potential density anomaly and the threshold value used to estimate the SML base depth.

f. I suggest to use different colors for SML and CHL segments. If they occupy the same depth, one can barely separate them so both are clearly seen.

In addition to the different line styles, we now also use two different colors. We chose fairly dark colors because of their high contrast compared to the background.

g. Overall figure structure: Instead of "ITP74," I suggest to use general title "Methods" and subtitles for the three right columns using methods' abbreviations.

Done.

h. Caption: please define stability (and its unit). "threshold values," "temperature difference" should be briefly explained.

Stability ($L$, unitless) was defined in Sect. 2.1.3. We added "($L$, unitless, see Sect. 2.1.3)" to the caption of Figure 2 instead of Figure 3, because this is where stability is first shown in a figure.

23. Lines 219-222: The sentence needs editing.

We shortened the sentences:

*This appears to be consistent with the mechanism for halocline formation described by Rudels et al. (1996). As stated above, Rudels et al. (1996) found new halocline formation taking place when relatively fresh shelf waters near the surface were advected above denser and saltier water below, limiting winter convection, while Rudels et al. (2004) and Alkire et al. (2017) stressed the role of melt water in general (including non-shelf water) in the warm Atlantic inflow through the Fram Strait and the Barents Sea for halocline formation via this type of capping mechanism.*

to:

*This appears to be consistent with the convective homogenization and capping mechanism for halocline formation described by Rudels et al. (1996).*

24. Line 222: This sentence needs an explanation helping to see that "the convection affected halocline water" (e.g. ..."as expressed by ..."). There are many sentences throughout the text like that (e.g., lines 228, 229-230, etc.). The text will benefit from clarifying them.

We softened the statement in line 222 saying "Figure 2 suggests that in this particular case, the convection may have affected halocline water" and added the explanation: "because prior to the onset of convection, Fig. 2 shows a well-defined halocline". We split the sentence previously starting in line 228 into two and explained that it is the identification of a stability maximum associated with fresh water near the surface that makes us think that the ST algorithm might capture the beginning of new

halocline formation (please refer to lines 226f of the revised manuscript or lines 255 of the track changes version).

25. Lines 236-237. This sentence ("Reasons..") can be deleted.

Done.

26. Lines 240-250 etc. The authors often use reference to figures inside the sentences (e.g., ll. 242 twice, 246). Traditionally, however, these references are placed at the end of the sentences.

We moved the references to the end of the sentences in this part and in a few other instances.

27. Lines: 249-254: I disagree that the differences between these methods are in different direction of search. This is a very misleading explanation. I think the major reason is, as the authors rightly stressed prior to this paragraph, a better physical justification, reasoning for the ST method. I suggest to remove this discussion or edit providing a better explanation.

In the revised version, we clarified that this particular statement pertains to the robustness of the methods. We also added a sentence to support our conclusion that the search direction helps to explain a drastic reduction in the number of artifacts in the ST method compared to the other methods. We are convinced that looking at the different search directions provides a key for understanding the increased robustness of the ST method compared to both other methods. Although we did not mention this in the first draft of our manuscript, during the development of the ST method, we specifically changed the search direction in the ST method in order to reduce artifacts associated with the spurious detection of halocline base depth minima. Conversely, reversing the search direction in the ST method results in spurious halocline base depth minima.

28. Line 259: "The underlying grid..." – this sentence can be removed.

Done.

29. Figure 6: I suggest to remove panels f-j and corresponding text referencing it. In the Arctic Ocean, there are no differences. For d-e: How the differences were computed if the points in (a-c) were not occupying the same positions?

A drawback of the DR method is that it identifies a halocline base outside the Arctic Ocean in the Norwegian Sea. This becomes especially problematic as similar conditions to the ones encountered in the Norwegian Sea today may extent further north in the future. We think that a correct halocline identification in regions which are prone to changes is very important if an algorithm is to be used in a climate change context. In our opinion, masking out regions solely based on location is not a good idea in a situation in which water masses may change. This point is further explained in our answer to comment #32 below.

30. Lines 265-280: Please do not use negative definitions in describing the ST method. E.g. Line 266 states that the ST "overestimates" halocline depth compared with another method. That really reads as negative towards the ST.

The "overestimate" in lines 265–280 referred to the TD method. We replaced

*while the TD method overestimates halocline depth relative to both other methods (Fig. 6b)*

by:

*while the TD method placed the halocline at greater depth compared to both other methods (Fig. 6b)*

The statement:

*This overestimate of the halocline base depth by the TD method compared to the other two methods is consistent ...*

was adapted as follows:

*This greater halocline base depth from the TD method compared to the other two methods is consistent ...*

(Although we are slightly concerned that now this sounds like something "great" for the TD method).

The sentence from Sect. 3.2:

*The more frequent halocline base depths larger than 120 m in the ST method are likely related to an overestimate of halocline base depth similar to the one found for the ST method for ITP-33 above.*

was changed to:

*The more frequent halocline base depths larger than 120 m in the ST method are likely related to a deeper halocline base similar to the one found for the ST method for ITP-33 above.*

Finally, the sentence from Sect. 4:

*In the Canada Basin, the new method overestimated the halocline base depth compared to the DR method, which correctly identified the halocline base for ITP-33.*

was adopted as follows:

*In the Canada Basin, the new method placed the halocline base deeper than the DR method, which correctly identified the halocline base for ITP-33.*

31. Same text: I suggest you drop off the definition of regions using ovals. You may refer to the regions using their geographical names.

    The reason for mentioning the elliptical regions is that we computed the summary statistics for the ellipses covering the regions.

32. Bottom of page 13: I strongly suggest that the authors drop off any mentioning of the Norwegian Sea since this is a very different region with very different water masses and driving forces. They can mask this area in Figs.

    One of our goals was to design a robust method for detecting the cold halocline base which can be used in a climate change context in those regions which are most likely affected by climate change, and which may

actually shift as the Arctic either warms or cools. In a previous study, we found large differences in the Norwegian Sea and also the Greenland Sea between the DR and the TD method when comparing the frequency of halocline thinning events between climate model results for past and near present-day conditions. In our opinion, masking out regions solely based on location is not a good choice when investigating climate change because conditions are not stationary. We are especially concerned that conditions which were previously encountered only in the regions which were masked out may with time propagate into the part which initially was not masked out.

We added the following sentences to the first paragraph of Section 4:

*Furthermore, the new ST algorithm does not require masking out predefined geographical regions based solely on their location. This is important because in a climate change context, the location of water masses may shift as the Arctic warms or cools.*

33. Figure 7: Instead of "relative frequency" the authors may use "histogram".

We have a slight preference for "relative frequencies" because a histogram can either show frequencies or relative frequencies.

34. Section 3.3: Lines 312-319 is a repetition of Intro and should be removed. Lines 319-323 belong to "Methods".

We removed the sentence

*The upper vertical stability maximum is associated with an increase of salinity near the top of the PWW and the lower stability maximum is associated with an increase in salinity between PWW and LHW.*

and also the sentence

*The new algorithm described in Sect. 2.2 was designed to provide estimates for the location of the cold halostad layer boundaries and the center, where the center is assumed to be the mean depth between the upper and the lower boundary.*

However, we retained other sentences which do not merely repeat the introduction, but instead refer to and explain the profiles in Figure 5. We also retained the sentences which explain the occasional gaps in the green line in Figure 4b–d, although they indeed do repeat points previously explained in the methods section.

35. Line 332: Please direct the readers where exactly they can see the cold halostad near the coasts of Greenland in Fig. 8b.

Thank you very much. We now direct the readers to Figures 8a and b.

36. Lines 329-332: Fig 8c may benefit from showing Standard Errors – they may help address the problem stated in the text.

Yes, thank you very much. Instead of standard errors, we show the 95% confidence interval of the mean (roughly twice the standard error) in the revised version of Figure 8c. The relatively narrow confidence intervals suggest that the difference between August, September, and October and the other month cannot be explained by different spatial sampling with more points at the edge of the region in August, September, and October. We deleted this suggestion from Section 3.3. We also omitted the red line showing the fraction of grid points for which at least one observation was available in the respective month in the revised version of Figure 8c.

**Response to reviewer comments by Marylou Athanase**

Thank you very much for your very insightful second review and very constructive comments. We repeat your comments (using a dark blue font color) below. Our responses are given in black font.

This is my second round of reviews for this manuscript. In my first review of the original manuscript, my main concerns regarded (i) the reasoning being the choice of the ST criteria, (ii) the lack of direct evaluation of the methods, and (iii) improving the manuscript's readability. The authors have answered all my initial comments, and I find that the revised manuscript addresses well these three main concerns. The introduction and method description were greatly improved. The result section is now clearer and presented in a more linear way. I appreciate the addition of Figures 3, 5, and 7.

I still have a few remarks regarding editing and text clarifications, and one main comment which I trust the authors will be able to address. Overall, I believe the manuscript is close to being fit for publication, and would recommend acceptance after some minor revisions.

**Main comment:**

Since the choice of the DR criterion used for computing the ST criterion is, to some extent, somewhat arbitrary, it would be nice to add 1 or 2 sentences discussing this choice & how it should be optimized in the last section of the manuscript. You already touched on this topic by indicating that the ST criterion could be slightly changed to provide a better match with the DR criterion. This raises two questions:

- Should one aim for a match between both methods?

- If yes, are there particular reasons or limitations to why the adjustment of the ST criterion, following your own recommendations, could not be included in this manuscript as well? Adding an attempted "optimal ST criterion" for future users could strengthen your manuscript even more.

We think that one should not per se aim for a perfect match between the two methods. However, in the future it would probably be good to determine an ST threshold based on an empirical analysis of profiles, similar to how Bourgain and Gascard originally determined the DR threshold. We added the following sentences to the second paragraph of Sect. 4:

*However, instead of simply aiming for a better match between the ST and the DR method, it may ultimately be more desirable to further fine tune the ST threshold based on an empirical analysis of profiles, similar to how Bourgain and Gascard (2011) determined the DR threshold. On the whole, estimating the ST threshold based on the DR threshold as we did provides fairly reasonable results, even without further fine tuning.*

The second sentence is meant to discuss and defend our choice without downplaying the potential for improvement. Although we did not actually test using different ST thresholds, we do not expect major changes in the overall outcome, apart from probably a somewhat less deep halocline base in the Canadian Basin. In particular, we do not expect that moderate changes of the ST threshold would reduce the robustness of the ST method. Regarding new halocline formation, we may have to look for a different case to demonstrate that in principle the capping part is captured by the ST method. But again, we do not expect the overall conclusion to be affected by the exact choice of the threshold. In hindsight, we should have considered basing the ST threshold more on the empirical analysis of profiles and considered abandoning our idea of establishing a link with the well-established DR method.

**Editing remarks:**

L. 6 and 338: Could you clarify what is meant by "robust" here? Most likely is done in the core text, but if mentioned in the abstract it should be clear what the criteria are for readers.

In line 6, we omitted the sentence that explicitly stated our intention to develop a method that is more robust based on a suggestion by I. Polyakov. We retained the next sentence in which we connect "robust" with "few artifacts". In line 338 we added "with few artifacts" in order to explain the meaning of "robust".

L. 28-30: This is confusing: you first refer to the advection of shelf water onto more saline waters as a halocline formation mechanism. Yet in the following sentences, you state that this mechanism is supported by 2 other studies, but then cite the formation of halocline via ice melting on top of warm AW. Although there are similarities in these mechanisms, they are not exactly the same. In the first, the input of freshwater comes from the fresher shelf waters,

themselves fed by continental runoff and/or ice melt. In the latter, only direct ice melt on top of saltier waters is at play. Please clarify this statement.

Thank you very much. Based on a comments by both reviewers, we modified our discussion regarding the the interplay between convective homogenization and capping in the introduction of the manuscript. The revised section reads:

*Based on data from the Oden 1991 cruise, Rudels et al. (1996) found that new halocline formation was initiated by the advection of relatively fresh shelf waters near the surface above denser and more saline water below, when the advection of the fresh water limited winter convection. Alkire et al. (2017) and Rudels et al. (2004) argued that convective homogenization and capping by fresh water due to sea ice melting in the inflow from the Fram Strait and the Barents Sea can transform AW into halocline water. Rudels et al. (2004) stressed that melting provides a precondition for sea ice formation and convective homogenization during the following winter.*

We decided to omit the following sentence, which we deem less relevant in the light of our extended discussion on the roles of melt water:

*Steele and Boyd (1998) argued that seasonal capping by melt water in summer may not be overly important for insulating the SML from relatively warm AW.*

Throughout the manuscript, we now refer to the mechanisms that involve convective homogenization and capping as "convective homogenization and capping" instead of simply "capping".

L. 40: This is again confusing. I understand what you mean, but calling the PWW both a part of the halocline and a halostad layer can appear contradictory. You could consider saying that the halocline in the Canada Basin is constituted of PHW, in which we distinguish a halocline layer formed by the PSW, which overlays a cold halostad formed by PWW, and below the PHW is found the strongly stratified LHW of Atlantic-origin. Adding a schematic figure of what these layers look like in a T-S profile could also help. You could point to the local stability maxima as well.

We omitted the reference to the statement by Zhong et al. (2019) that the PWW could be considered a type of cold halocline water. Regarding the local stability maxima, we refer to Supplement 1, which provides a schematic of the salinity and the stability profile and to Supplement 2 which shows profiles

based on observations. The salinity profile in the schematic is qualitatively similar not only to the profiles in Supplement 2, but also to the profiles shown in figure 2 of Shimada et al. (2005). The salinity and stability profiles in the schematic are also qualitatively similar to the corresponding profiles in Figure 5a and d except for the layer of Pacific Summer Water. They are also qualitatively similar to the corresponding profiles in Figure 5e and h except for the SML.

L. 50: typo "ist"

Thank you. We deleted the "t", translating the German word "ist" (meaning "is") into English.

L. 59: This sentence should rather be merged with line 52, here it is a little off-topic.

This stand-alone sentence is indeed awkward. However, we think that moving the sentence to line 52 would break the logic around line 52.

L. 67: "among others by others" typo

Corrected. Thank you.

L. 86-87: This sentence should be moved to the end of the previous paragraph (l. 59), as it results from emphasizing the importance of the halocline in the Arctic system.

We moved the sentence to the suggested paragraph, and linked it adding "Therefore," at the beginning.

L.88: The manuscript outline is long and too detailed. Consider shortening this part, to no more than 1 sentence per section (1-2 lines) without detailing each sub-section. You might also not need to acknowledge your reviewers' suggestions.

We shortened the section and cut the references to the reviews based in part also on a suggestion by I. Polyakov. The reviewers' suggestions are still acknowledged in the acknowledgment section. But we did like the idea of citing

the reviews in the text, and had initially found numerous arguments to defend our choice of citing the reviews.

L. 148: "as demonstrated below" What does this refer to? This section? A figure?

We now refer to Supplements 1 and 2. Originally, we meant Fig. 5e and f.

Fig. 3: Nice figure. - tick labels are occasionally overlaying each other, consider fixing that - panel label and plot lines are occasionally overlaying each other, consider fixing that - if possible, would be great to have a profile of density as well.

We fixed most overlaps. Figures 3 and 5 of the revised manuscript include profiles of potential density.

L. 263-310. This section is interesting but quite dense. Consider breaking it down into several paragraphs, e.g., by adding a line break l. 271 (before "In addition") and l. 283 (when transitioning to the Canada Basin).

Done. Thank you.

L. 275-277: Somewhat convoluted phrasing. In general, it is easier when sentences are straight to the point. In this case, you could just say that "the ST method also detects more frequent cases of halocline BD below 120 m compared to the DR method."

Thank you very much. We changed

*More frequent detections of halocline bases not only above 60 m but also below 120 m with the ST method compared to the DR method account for a slightly wider halocline base depth distribution in the Eurasian Basin in the ST method compared to the DR method (Fig. 7b).*

to

*The ST method also detects more frequent cases of halocline base depth below 120 m compared to the DR method (Fig. 7b).*

L. 282: Any comments on this? You don't make any use of Fig. 7c otherwise.

We now explicitly point out that the TD method "again" yields a greater halocline base depth compared to the other two methods, where the "again" refers to the base depth in the Eurasian Basin. This helps to highlight the contrast to the Canadian Basin, which is discussed starting with the next sentence.

L. 312-316: This is merely a suggestion, and you may choose or not to follow it: since you have already explained your concepts in section 2, you can consider removing the reminder of these concepts at the beginning of your subsections in section 3. Here, for example, lines 312-318 mostly repeat information provided in section 2. Since the paper is not overly long, you can however decide to maintain these reminders if you prefer to do so.

We removed some of the reminders (in part also based on a comment by I. Polyakov), but kept the ones which directly refer to the results in Fig. 5. We also removed a repetition from the method section in the same paragraph based on suggestion by I. Polyakov, but kept the repetitions that we use to explain the occasional gaps in the green line in Fig. 4.

L. 335: Redefine all acronyms in the summary section

Done.

L. 374: This is a bit colloquial (and rather pessimistic wording). Consider saying rather that e.g., "mismatch rates seem to be notably varying across ITPs tested. This suggests that further testing and refining are needed in order to account for regional and temporal variability of CHS characteristics"

Thank you very much. We reworded

*while some other ITPs yielded almost perfect results and some other ITPs slightly worse results*

to

*albeit with detection rates varying across ITPs tested from almost perfect to a slightly higher mismatch rate*

However, we retained our (more optimistic?) outlook. Although the method is not perfect, we do think that it may already be useful in its present form. In order to make this clearer, we modified the sentence:

*This suggests that a stability-based method for CHS detection could be useful for future studies exploring variability and changes of the CHS in the Canada Basin.*

to:

*This suggests that the new stability-based method for CHS detection could be useful for future studies exploring variability and changes of the CHS in the Canada Basin.*

L. 381: I would suggest rephrasing your very last sentence so that it highlights the potential of AI methods without discarding too harshly traditional methods (including your newly proposed one).

We removed the sentence

*Given the various shortcomings of traditional threshold methods, AI-based methods could nevertheless be useful.*

and instead speculate:

*Rapid advances in AI may help to overcome these problems.*

The original sentence was based on the (pessimistic) speculation that at least in the near future neither of the methods is likely to function perfectly, and on the (optimistic) speculation that in the meantime both methods may be useful in spite of their shortcomings.

**Response to editor comment by Ilker Fer**

When preparing your revision, could you also replace "Mai" with "May" (Fig 3e), and, if possible, make Figs7-8 50% or so taller?

Thank you very much. We replaced "Mai" with "May" in Fig. 3e, made the profile plots in Fig. 7 taller, and also enlarged and rearranged the panels in Fig. 8.

*Supplement of*

**Technical note: Determining Arctic Ocean halocline and cold halostad depths based on vertical stability**

**Enrico P. Metzner and Marc Salzmann**

*Correspondence to:* Enrico Metzner (enrico.metzner@uni-leipzig.de)

- supplement-title-page.pdf
- supplement1.pdf
- supplement2.pdf
- supplement3.pdf
- supplement4.pdf
- supplement5.pdf
- supplement6.pdf

**Supplement 1**

[Figure]

**Figure S1.** Schematic salinity (a) and stability (b) profile in the Canada Basin. Blue shadings in (a) indicate an increase in salinity below the surface mixed layer (SML) associated with the Pacific Halocline Water (PHW) and another increase associated with the Lower Halocline Water (LHW) below the PHW. These salinity gradients result in two distinct local stability maxima between the surface and the Atlantic Water below the LHW (b). The salinity profiles in (a) are similar to the ones in Shimada et al. (2005). Supplement 2 shows similar profiles for salinity and stability from ITP-41 in the Canada Basin and also includes temperature profiles.

**Supplement 2**

[Figure]

**Figure S2.** Profiles of salinity (blue), temperature (red), and stability
(brown) from ITP-41 Level III data for days from 16 Feb 2011 to
27 May 2011 (Krishfield et al., 2008; Toole et al., 2011). Compare
also schematic in Supplement 1. Salinity is in the practical
salinity scale. Stability ($L$, unitless) was computed as described
in Sect. 2.1.3. Below  280 m, thermohaline staircases are found in
the Atlantic Water.

**Supplement 3**

[Figure]

**Figure S3.** Example: determining the CHS boundaries and center depth (see Sect. 2.2) based on the stability profile for ITP-33 on 6 September 2010. Corresponding temperature and salinity profiles are shown in Fig. 5f. Please refer to the text below for an explanation of the figure.

First, the base of the halocline is estimated using the stability algorithm as described in Sect. 2.1.3 **(a)**. The thin vertical blue dashed line in (a) indicates the stability threshold for estimating the halocline base (searching upward). The horizontal thick blue dashed line indicates the halocline base and the dotted gray line the SML base (computed as described in Sect. 2.3). Then, between the SML base and the halcoline base, a moving maximum is computed for the stability as described in Sect. 2.2 (brown line in **(b)**). The first condition for detecting a CHS is met if a minimum is found above the halocline base between two stability maxima based on the moving maximum profile (red circle). This implies that the distance between the deeper stability maximum and the first upper occurrence of that same stability value must be at least $50\,\mathrm{m}$. An additional condition for identifying the CHS is that the difference of the stability $L$ between the lower stability maximum and the local minimum (i.e. the "amplitude" of the minimum) in the CHS is at least 0.2. If these condition are met, we compute the mean (thick dashed vertical green line in **(c)**) of the stability minimum and the deeper stability maximum (thin dashed vertical green lines in (c)). This stability value is used as a threshold to identify the upper and the lower bound of the CHS (horizontal solid green lines in (c)). The center depth of the CHS is defined as the mean depth between the upper and the lower bound (not shown here). The conditions regarding the minimum vertical extent and the amplitude of the stability minimum described above help to minimize the number of false CHS detections.

**Supplement 4**

[Figure]

**Figure S4.** As Fig. 3 but based on Level I data and using the International
Thermodynamic Equation of Seawater – 2010 (TEOS2010,
McDougall et al., 2010) to compute potential density, potential
temperature, and potential freezing temperature instead of Gill
(1982).

**Supplement 5**

[Figure]

**Figure S5.** ITP-74 location of measurements (indicated by small red crosses along the tracks) and sea ice concentration (area fraction) from Spreen et al. (2008) for the profiles shown in Figure 3. Sea ice data was obtained from `https://www.meereisportal.de` (Grosfeld et al., 2016).

**Supplement 6**

[Figure]

**Figure S6.** As Fig. 5 but based on Level I data and using the International Thermodynamic Equation of Seawater – 2010 (TEOS2010, McDougall et al., 2010) to compute potential density, potential temperature, and potential freezing temperature instead of Gill (1982).

[revised manuscript text omitted]
 [M.A.]surface mixed layerSML from warmer [M.A.]Atlantic Water (AW[M.A.]) below and thus acts

[Figure]

(a) CHS center

(b) CHS thickness

(c) RF$_{CHS}$

**Figure 8.** Please note: Figure 8c was revised to show 95% confidence intervals for the mean. The fraction of grid points for which at least one observation was available in the respective month was omitted. The panels were enlarged and rearranged. The color bars on the maps are now oriented horizontally instead of vertically. Maps of mean  CHS center depth (a) and  CHS thickness (b).  Monthly relative occurrence frequency (RF$_{CHS}$[I.P], blue bars) of the CHS in the Canada Basin region (area enclosed by red dashed lines in [I.P]b) and 95% confidence interval for the mean (c)(b) .

375  to protect sea ice. Another objective was to design a particularly robust method [M.A.]with few artifacts. [I.P]Furthermore, the new ST algorithm does not require masking out pre-defined geographical regions solely based on their location. This is important because in a climate change context, the location of water masses may shift as the Arctic warms or cools. We also devised a new stability-based method to identify the [M.A.]cold halostad (CHS[M.A.]), which is formed by [M.A.]Pacific Winter Water (PWW[M.A.])

[revised manuscript text omitted]

---

## Author Response (AR3)

**Response to reviewer comments by Igor Polyakov**

Thank you very much for reviewing our revised manuscript. We repeat your review (using a dark red font color) below. Our response is given in black font.

In this review, I limited my task by checking only the authors' response to my major point of concern, namely excitation of spurious salinity extremes in high-gradient segments of the profiles. This is a well-known problem based on different sensitivity and time constants of temperature and conductivity sensors. I am not happy with the authors' response to my comment. They argued that they noticed the same issue in modeling data. This argument is left to the conscience of those who processed the modeling data. The authors also suggested that PWW could be one of the reasons for this spiritual salinity extremes. These extremes, however, are visible in ITP74 salinity profiles. I am not here to argue that salinity profiles can exhibit false extremes in areas with significant T&S gradients. There is an abundance of literature on this subject. This is a serious problem, and because of the nature of the task in front of the authors, it should be addressed. Right now, the issue is not handled properly. In its turn, this issue may affect the authors' results (e.g., the DR method may select density values affected by the spurious salinity extremes). The optimum solution would be to reprocess the data and to use lagged correlations inside a running window to locally adjust the T and S profiles. However, due to the relatively coarse ITP resolution, it may be prohibitively time-consuming and may not totally fix the problem. Another (easier) technique would be to apply extra smoothing (the authors did so anyway), but attention should be given to explain the overall impact of this smoothing. The problem must be resolved in some way. Otherwise, this potentially extremely important work would leave holes in the authors' conclusions. I leave the decision to the editor, but I would advise the authors to take this issue carefully.

We thank the reviewer for stressing this issue. Based on the recommendation by the Editor, we now alert readers to this problem (for details please refer to our response the the Editor). Undeniably, we lack the expertise and the

funding to reprocess the raw data ourselves. As far as we understand, much effort has already been directed at addressing this well-known problem, making further improvements rather challenging. We did smooth the observations in order to reduce the noise, and initially also tried different filters, but at the time we did not find major improvements over the filter that we eventually chose (not shown). Regarding the model data, we would like to stress that model data does not suffer from issues related to sensor sensitivity. But there may be other issues, and while higher-order numerical schemes produce unrealistic extrema in the vicinity of steep gradients, much effort has been spent on developing monotonic schemes. Furthermore, when analyzing time averaged model output, such potential artifacts (which can occur for non-monotonic advection schemes) would be expected to average out, especially as the averaging time increases. Although one should in principle use instantaneous model output for this analysis, such output is not always available. Therefore, our analysis in Metzner et. al (2020) was based on time average model output, which is expected to efficiently smooth out numerical artifacts. (Furthermore, we were very careful to check that the overall results by Metzner et al. were plausible in the light of the simulated temperatures and salinities, because we noticed artifacts in the halocline base detection as previously stated). When applied to the widely used Level III data as described in the manuscript, the ST method yielded less artifacts compared to the DR method in the halocline base depth. We expect this result to be robust and we hope that the ST method turns out to be useful.

**Response to reviewer comments by Marylou Athanase**

Thank you very much for reviewing the manuscript again and for your editing remarks. We repeat your comments (using a dark blue font color) below. Our responses are given in black font.

This is my third round of reviews for this manuscript. In my previous round of review, I had only one main comment regarding the DR criterion used for computing the ST criterion. The authors explained their reasoning in their response, and address this briefly but clearly in the discussion. The added figures in the supplementary are also a nice addition. My other minor comments were addressed, and I find that the authors have satisfactorily answered the other reviewer's concerns.

I believe the manuscript is now fit for publication. Below are some small editing remarks, that the authors may choose or not to follow.

Editing remarks:

L. 4-5: I would suggest you rather explain what the cold halostad is (i.e., a layer of locally minimal vertical salinity gradient). It is sufficient to introduce the water sources in the main text, as you do.

We added the explanation "a layer characterized by a small vertical salinity gradient", but retained the part of the sentence explaining the water sources.

L. 6-7: "Unlike...": I suggest rephrasing by something along the lines of: "Vertical stability is a crucial parameter for determining whether the halocline can prevent vertical heat exchange and protect sea ice from warm AW, which is less/not captured by previously used parameters."

We deleted "Unlike parameters previously used to detect the halocline base" form the sentence. In the main text, we had already explained that "[t]he choice of a vertical stability threshold was motivated by the argument that vertical stability is more directly related to vertical mixing than either density, temperature, or the density ratio." We previously found a statistical correlation between changes of halocline properties, heat fluxes, and sea ice cover in climate model data. We argued this correlation reflected a causal link based among others on seasonality (Metzner et al., 2020). A cold halocline base diagnostic that is per construction more closely related to vertical mixing will help to strengthen this argument.

L. 172-173: Please rephrase the sentence into: "The accuracy of the sensors used for the ITP observations is 0.002 ∘ C for temperature and 0.002 for salinity according to the manufacturer (Janzen et al., 2016)."

Thank you. Done.

L. 327-328: I still don't know which points are supposed to be the CHS detected near Greenland: do you means the few detected grid cells to the northeast of Greenland, facing Svalbard from the west side of Fram Strait? If yes, please add these geographical indications to your text and/or figure.

Yes, we meant these points. We replaced "near the coasts of Greenland" by "to the northeast of Greenland, facing Svalbard from the west side of Fram Strait". We also softened "where glacial cold water acts similar to the Pacific low salinity water (Dmitrenko et al., 2017)" to "where glacial cold water may act similar to the Pacific low salinity water (Dmitrenko et al., 2017)".

L. 335-337: This would be a good place to mention the ongoing Atlantification and associated northward shift of ocean conditions previously specific to Nordic Seas / marginal regions.

We added "An example is the ongoing atlantification and associated northward shift of ocean conditions previously specific to Nordic Seas / marginal regions (e.g. Athanase et al., 2021)."

**Response to editor comment by Ilker Fer**

Thank you very much for your editor report. We repeat your report (using a dark green font color) below. Our response is given in black font.

Thank you for your thorough revision of the manuscript.

Please address the minor issues pointed by Dr. Athanase.

Regarding Dr. Polyakov's comment: this is now archived in the discussion and the reader is alerted about this potential issue. In your analysis, you are using the Level 3 ITP data, which applies data processing including all sensor corrections in the best possible way. It will also be the choice of potential users of your proposed method. In addition, you apply vertical smoothing which could remedy the issue Dr. Polyakov raised. Further detailed and advanced processing (with revised time constants or lags between sensors) is not critical. However, I request that you explicitly mention this caveat in the summary and conclusions (also earlier in the discussion of results where relevant) along the lines of "... observations show spurious salinity extremes in high-gradient segments which can have consequences in the application of the methods (e.g., the DR method may select density values affected by the spurious salinity extremes), and caution is advised in the application of the method." Please improve this text as needed.

We addressed the minor comments by Dr. Athanase as described above.

Regarding the Dr. Polyakov's comment, we added statements to the discussion of results and to the summary and conclusions section:

Because Dr. Polyakov stated that Figure 3 was his major concern, we added the following sentences to the discussion of Figure 3:

*A caveat is that observations are known to show spurious salinity extremes in high-gradient segments because of different sensitivity and time constants of temperature and conductivity sensors (e.g. Johnson et al., 2007). This can have consequences in the application of the methods. For example, the DR method may select density ratio values affected by the spurious salinity extremes. Although here Level 3 ITP data was used, which is processed data applying sensor corrections, it is possible that salinity profiles exhibit false extremes in areas with significant temperature and salinity gradients.*

Here, we added the citation to Johnson et al. (2007). We chose this reference because we think that it describes the well-known problem Dr. Polyakov has been referring to and because it contains several references to previous publications on this issue already in the first paragraph of the introduction. The SBE-41CP CTDs discussed by Johnson et al. (2007) sensor is used in the Ice-Tethered Profilers. According to Krishfield et al. `http://www.whoi.edu/fileserver.do?id=35803&pt=2&p=41486`, corrections suggested by Johnson et al. (2007) were applied to Level 3 data.

We also included a note of caution in the summary and conclusions section. It reads as follows:

*A caveat potentially affecting the halocline base depth determined via the DR and the ST method is that observations show spurious salinity extremes in high-gradient segments which can affect the outcomes. The DR and the ST method may select density values affected by the spurious salinity extremes. Therefore, caution is advised in the application of the DR and the ST method to observations.*

We corrected the access date for the ITP-data in the code and data availability section from 23 January 2022 to 23 January 2023.

**References**

[revised manuscript text omitted]